# Substorm expansion embedded in a global cycle of field-aligned currents and auroral electrojets

Tonghui Wang[1,2], Lei Dai [1] ✉, C. Philippe Escoubet[3], Walter Gonzalez[4], Yong Ren[1], Minghui Zhu [1], Shan Wang [5], Chi Wang [1], Xu Wang[1,2], Kailai Wang[1,2] & Jinjuan Liu[6]

Geomagnetic substorms transfer solar wind energy into the planetary magnetosphere and ionosphere, producing auroral displays and ground magnetic disturbances, particularly intense during the expansion phase. Despite decades of study, the mechanisms governing the expansion phase remain unresolved. Based on coordinated observations of storm-time intense substorms, we reveal that substorm expansion is temporally embedded within a global cycle of field-aligned currents and auroral electrojets, coupled to large-scale plasma convection. The cycle manifests as a coherent movement of current peaks across magnetic longitude and latitude—first antisunward and equatorward, then sunward and poleward—and coincides with enhanced sunward ionospheric convection. This cycle involves two components of the auroral electrojets: the convection-driven DP-2 current and the expansion-phase DP-1 substorm current. The antisunward-equatorward phase, corresponding to intervals of dominant dayside reconnection, begins with DP-2 and can stepwise transition into DP-1. During the subsequent sunward-poleward phase, reflecting intervals of dominant nightside reconnection, DP-1 either persists from the earlier interval or develops within this phase. These observations show that expansion onset can occur under dominance of either dayside or nightside reconnection, while the full development of DP-1 generally involves nightside reconnection, providing insight into substorm evolution.

Substorms are among the most dynamic manifestations of solar-terrestrial coupling, marked by intense auroral displays, large-scale reconfigurations of electric current, and intense magnetic disturbances in the geospace[1–3]. They occur not only in terrestrial space but also in planetary environments[4,5]. Substorms proceed through a three-phase cycle: solar wind energy is first accumulated in the planetary magnetosphere (growth phase), then abruptly released into the magnetosphere-ionosphere system (expansion phase), followed by a gradual relaxation of the magnetosphere-ionosphere system (recovery phase). Understanding substorm evolution is a central scientific objective of the Solar Wind Magnetosphere Ionosphere Link Explorer (SMILE) mission[6,7].

Because the expansion phase marks the rapid release of stored energy, its underlying mechanism has been the focus of decades of

[1]State Key Laboratory of Space Weather, National Space Science Center, Chinese Academy of Sciences, Beijing, China. [2]University of Chinese Academy of Sciences, Chinese Academy of Sciences, Beijing, China. [3]European Space Research and Technology Centre, European Space Agency (ESA), Noordwijk, Netherlands. [4]National Institute for Space Research (INPE), São José dos Campos, São Paulo, Brazil. [5]Institute of Space Physics and Applied Technology, Peking University, Beijing, China. [6]CMA-USTC Laboratory of Fengyun Remote Sensing, University of Science and Technology of China, Hefei, China. ✉e-mail: ldai@spaceweather.ac.cn

research. Previous research has focused primarily on identifying onset mechanisms that initiate substorm expansion, including near-Earth reconnection and/or plasma instabilities[8–12], as well as solar-wind-driven triggers such as interplanetary shocks, changes in Interplanetary Magnetic Field (IMF) $B_z$, and dynamic pressure variations[13–19]. While the debate over substorm onset continues, much less is known about the global processes that organize the full evolution of the expansion phase.

A key to understanding the full substorm evolution lies in the behavior of large-scale ionospheric and field-aligned currents (FACs). These include the eastward and westward auroral electrojets (AEJs) in the auroral-zone ionosphere and the FACs coupling the ionosphere to the magnetosphere. The AEJ is commonly interpreted as the superposition of two components: DP-1 and DP-2[20–22]. DP-1 represents the ionospheric part of the expansion-phase substorm current wedge[3,23–25]. Its rapid intensification—driven by enhanced conductivity from auroral particle precipitation—marks the expansion phase[22]. The DP-1 electrojet forms on the nightside, typically spanning magnetic local time (MLT) 21-04, and dominates during the expansion and early recovery phases[22].

In contrast, DP-2 is driven by large-scale magnetospheric convection throughout the entire substorm cycle[20–22,26]. This global convection is attributed to dayside reconnection[21] and progresses antisunward to the nightside[27,28]. The global magnetosphere convection couples to the ionosphere through large-scale FACs[28–34]. Global simulations show that convection-associated Region 1 FACs develop from the dayside toward the nightside[28,33–35], while observations reveal that global-scale FACs intensify and shift toward nightside during substorm activity[36–38]. Recent evidence further suggests that strong solar-wind-driven convection (DP-2) may directly induce the development of DP-1 under intense driving conditions[17].

Here we identify a global cyclic evolution of FACs and AEJs during a sequence of intense substorms in the 17 March 2015 geomagnetic storm. These cycles involve coherent motion of current peaks across magnetic longitude and latitude—first toward the nightside and equator, then reversing toward the dayside and pole. The accompanying latitudinal expansion-contraction is consistent with the expanding and contracting polar cap (ECPC) paradigm[39,40]. This correspondence indicates alternating dominance of dayside and nightside reconnection during each cycle. Together, these observations constrain the relative timing and development of DP-1 within the global reconnection process, revealing a key feature in the large-scale organization of substorm evolution.

## Results
### Cyclic evolution of FACs, convection, and westward AEJ
An overview of intense substorms during the 17 March 2015 geomagnetic storm is provided in Supplementary Fig. 1. Here, we focus on the first substorm to examine the detailed evolution of ionospheric convection, FACs, and the westward AEJ on the dawnside in Figs. 1 and 2. The corresponding IMF $B_z$ and SML index (the peak intensity of the westward auroral electrojet derived from SuperMAG[41–43]) are shown in Fig. 1a-b. During this substorm, the expansion phase is associated with intensified southward IMF $B_z$, while the recovery phase coincides with a subsequent northward turning of $B_z$. During this interval, comprehensive coverage across MLT and magnetic latitude (MLAT) on the dawnside was provided by ground-based magnetometer monitoring the AEJs and by a global radar network mapping ionospheric convection.

The Region 1 FACs, the westward AEJ, and the global convection pattern exhibit a coherent antisunward and equatorward progression during the expansion phase (Fig. 1c-d). The Region 1 FAC peak is initially located near MLT about 9-10 and MLAT about 70° at 12:40–12:50 UT. Over the next ten minutes, the peak shifts slightly antisunward and equatorward. More notably, the overall Region 1 pattern undergoes a substantial antisunward-equatorward

displacement due to the intensification of additional nightside FACs near MLT = 0-3, consistent with previous studies[44]. These additional FACs strengthen at these MLT location where the pre-existing Region 1 FAC have already extended. The global convection pattern evolves similarly, with a modest shift in the sunward flow peak, but a substantial movement of the overall pattern toward the nightside and equator. Region 1 FAC maxima typically appear at 60–70° MLAT, coinciding with the sunward return-flow region on closed field lines and therefore equatorward of the open-closed field boundary (OCB). The antisunward-equatorward motion is more pronounced in the westward AEJ. The SML peak shifts from MLT about 6 and MLAT about 65–70° at 12:40–12:50 UT to MLT about 3 and MLAT about 60–65° ten minutes later. This evolution reflects both the antisunward extension of the pre-existing AEJ near MLT = 6 and the emergence of additional nightside AEJ near MLT = 2.

The currents and convection progress sunward and poleward (Fig. 2a, b) In the early recovery phase. Between 13:30 and 13:40 UT, the Region 1 FAC maximum is located near MLT about 4–5 and MLAT about 65–68° and then moves to MLT about 7–9 and MLAT about 67–73° over the following twenty minutes. The sunward progression of Region 1 FAC involves a decay of the nightside FACs around MLT = 3–5 in the recovery phase, as consistent with that in ref. 44. In addition, the sunward extension of the pre-existing FAC to MLT = 6–9 also contribute to the sunward progression. This sunward progression reflects both the decay of the nightside FACs and the sunward extension of the pre-existing Region 1 system, again consistent with earlier work[44]. Convection follows the same trend: the sunward-flow peak and the overall pattern shift sunward and poleward, from MLT about 5–6 and MLAT about 65–67° to MLT about 7–8 and MLAT about 65–68°. The westward AEJ exhibits a similar clear evolution. The SML peak moves from MLT about 2–5 and MLAT about 60–65° at 13:30–13:40 UT to MLT about 3–6 and MLAT about 65–70° ten minutes later. This evolution reflects an sunward extension of the pre-existing AEJ to MLT > 6, and a decay of nightside AEJ near MLT = 2.

Supplementary Figs. 2 and 3 shows another cycle of auroral-current evolution between 15:00 and 17:00 UT, associated with the second and third substorms. From 15:10 to 15:50 UT, the peaks of the Region 1 FAC, ionospheric convection, and the westward auroral electrojet move antisunward and equatorward. The expansion phase of the second substorm occurs largely within this interval, consistent with the behavior of the first substorm in Figs. 1 and 2. For the third substorm, its expansion phase is mainly with the subsequent sunward and poleward motion of current peaks. This timing contrasts with the first substorm, in which the sunward-poleward motion occurred during the recovery phase. For the fifth cycle as in Supplementary Figs. 4 and 5, the antisunward and sunward evolution corresponds mostly with the growth phase and expansion phase of the sixth substorm, respectively. In this cycle, the motion of the current peaks appears mainly smooth, in contrast to the more stepwise progression observed during the first substorm (Figs. 1 and 2). A broader examination of timing relationships across all substorms is presented in the next section.

### Global MLT-MLAT cycles of Region 1 FACs and westward AEJs during consecutive substorms
We examine the temporal evolution of Region 1 FACs and the westward AEJ (represented by the SML index) across MLT and MLAT during six consecutive substorms (Fig. 3). Expansion phases, identified by sharp monotonic decreases in SML, are indicated by shaded vertical bands in Fig. 3a.

Panel (b) shows the regional SML as a function of MLT and time, with the MLT of the SML peak overplotted. To determine whether the peak AEJ corresponds to DP-1 or DP-2, we follow the spatio-temporal signal described in ref. 22. A SML peak is classified as DP-1 if (1) it occurs during the expansion or early recovery phase (operationally defined as

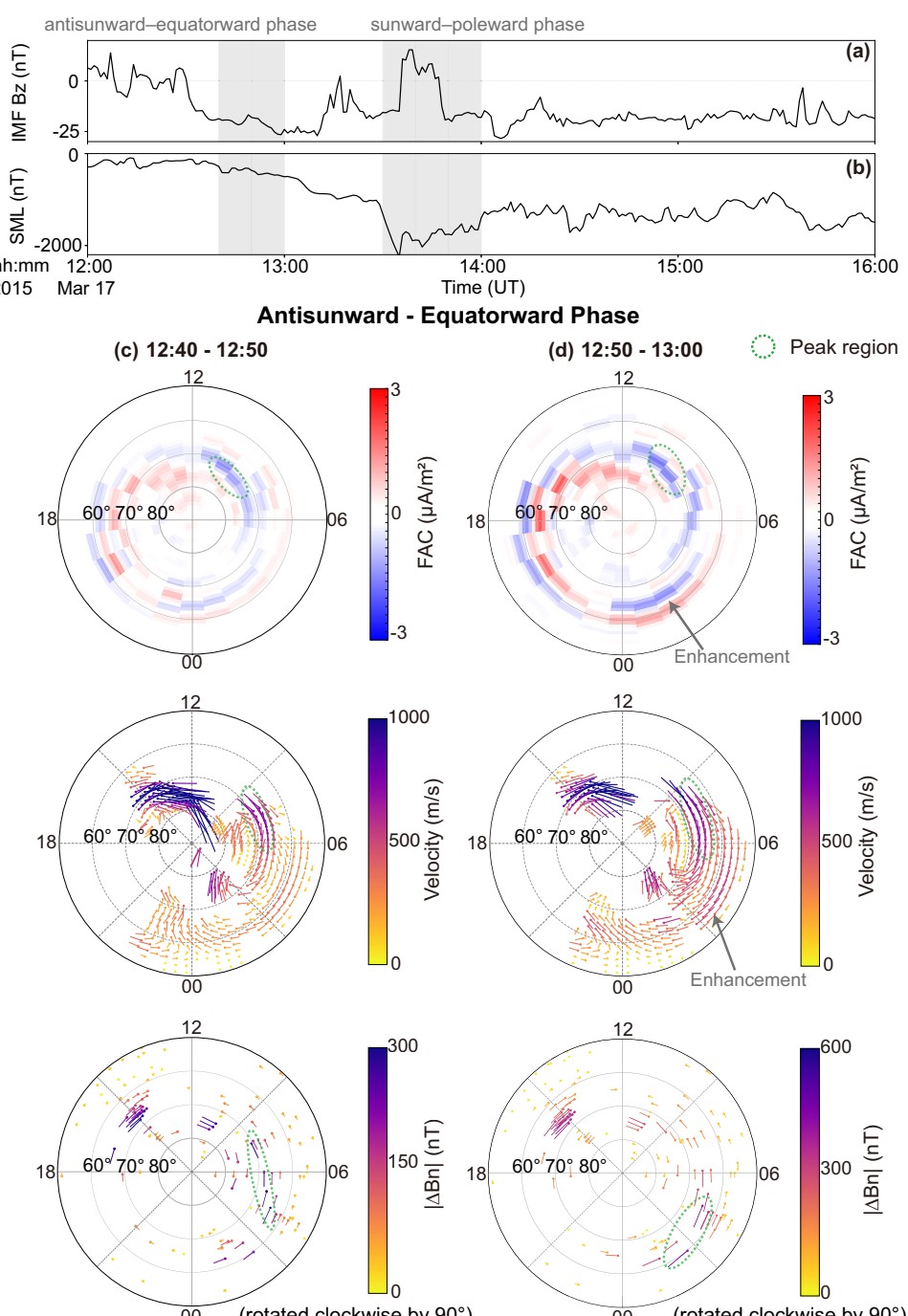

**Fig. 1 | Coordinated antisunward-equatorward evolution of Region 1 FACs, ionospheric convection and local SML during 12:40UT- 13:00UT on 17 March 2015. a** IMF $B_z$ and (**b**) SML index. The gray shadow mark time windows analyzed in Figs. 1 and 2. Snapshots of AMPERE-derived FACs (10-minute resolution), SuperDARN convection patterns (10-minute average), and local SML (10-minute median values of north-south $B_N$ perturbations rotated by 90 degrees, characterizing the aurora electrojet) during the antisunward-equatorward phase at selected intervals: (**c**) 12:40-12:50 UT, (**d**) 12:50-13:00 UT. Red and blue indicate upward and downward FACs, respectively. Dawnside Region 1 FACs are downward (blue). The green dashed lines highlight the peak regions of the data and grey arrows are used to emphasize data features. Source data are provided as a Source Data file.

$SML < 0.5*SML_{min}$), and (2) its MLT falls within the typical DP-1 sector (22-04 MLT), which can broaden during strong substorms[22,45] (e.g., the fourth substorm shows a DP-1 peak near 4.3 MLT). SML Peaks that do not satisfy these DP-1 conditions but coincide with enhanced ionospheric convection ( > 300 m/s; Fig. 3e) are classified as DP-2. DP-2 represents the large-scale convection-driven electrojet that persists throughout the substorm cycle but becomes masked when DP-1 intensifies.

Panels (c-d) reveal cyclic patterns in current peaks across MLT and MLAT in the dawn sector. Five distinct cycles are marked by grey boxes. These cycles are identified by examining the MLT peak of SML near the expansion phase. The antisunward phase is identified by a rapid decrease in MLT exceeding ΔMLT > 2 hours within 20 minutes, while a subsequent increase in MLT marks the sunward phase. The time windows derived from the MLT evolution are then applied to the MLat panels.

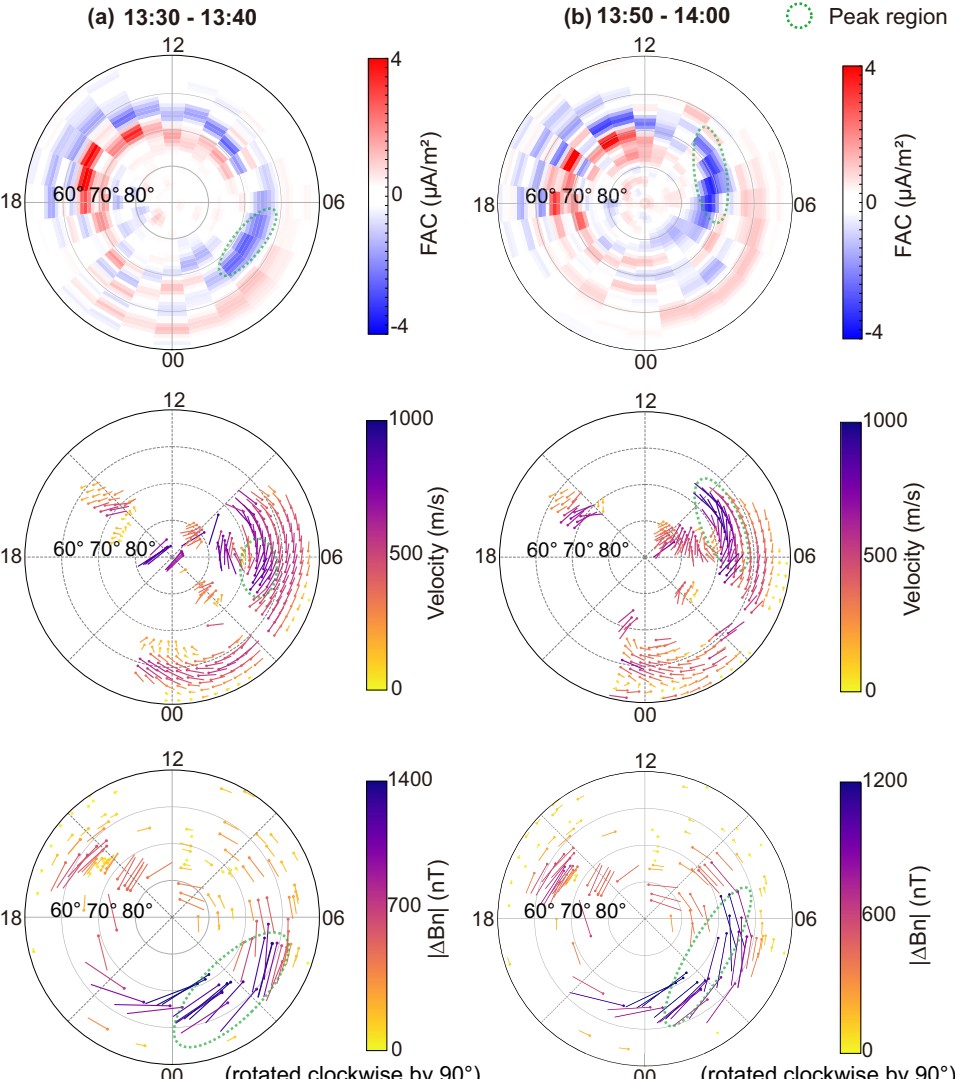

**Fig. 2 | Coordinated sunward-poleward evolution of Region 1 FACs, ionospheric convection and local SML during 13:30UT- 14:00UT.** Snapshots of AMPERE-derived FACs (10-minute resolution), SuperDARN convection patterns (10-minute average), and local SML (10-minute median values of north--south $B_N$ perturbations rotated by 90 degrees, characterizing the auroral electrojet) at selected intervals: (**a**) 13:30-13:40 UT, (**b**) 13:50-14:00 UT. Red and blue indicate upward and downward FACs, respectively. Dawnside Region 1 FACs are downward (blue). The green dashed lines highlight the peak regions of the data. Source data are provided as a Source Data file.

In the first cycle, the peaks of the westward AEJ (local SML) and Region 1 FAC peak moves antisunward and equatorward, forming a descending phase in 12:40-13:00 UT. From 13:20 to 14:10 UT, it reverses sunward and poleward, forming an ascending phase. This two-stage evolution is consistent with the two-dimensional pattern in Figs. 1 and 2. During this first cycle, the descending phase begins with DP-2 and then transitions into DP-1 during the expansion phase. DP-1 continues into the early recovery phase, extending into the ascending interval. Four additional MLT-MLAT cycles are observed, each lasting 40–140 min. In these later cycles, DP-2 either occupies the entire antisunward-equatorward phase (3rd–5th cycles) or transitions into DP-1 (2nd cycle). During the sunward-poleward phase, DP-1 either persists from the preceding interval (2nd cycle) or develops entirely within this phase (3rd–5th cycles).

A key observation is that all six substorm expansions are temporally embedded within the longitudinal-latitudinal cycles. The first two expansion occur largely during the antisunward-equatorward phase, whereas the remaining four occur mainly during the sunward-

poleward phases. All cycles coincide with strong dawn-sector ionospheric sunward convection (300–1000 km s$^{-1}$; Fig. 3e), indicating that the cyclic MLT-MLAT motion of the current peaks is closely tied to enhanced large-scale convection. The first two substorm expansion occurs under sustained southward IMF $B_z$ and positive $E_y$, and their recovery phases follow a northward turning of IMF $B_z$ (Fig. 3f). The 3rd-4th substorm expansion occurs in the decrease of $E_y$, indicating reduced dayside reconnection. The fifth and sixth substorm expansion occurs under nearly steady $E_y$ and southward IMF $B_z$. All four later substorms develop during intervals of strong ring-current enhancement (SYM-H about −100 to −200 nT), a condition under which the magnetotail is thought to become more resistant to reconnection[46,47]. This effect can contribute to the delayed onset of their expansion phases relative to intervals of strong solar-wind driving.

The substorms in our events share characteristics with the multiple intensifications[48] identified by the SOPHIE-M algorithm[49,50], and with global-convection events[51,52]. The distinction, however, is that multiple intensifications are typically marked by a lack of coherent

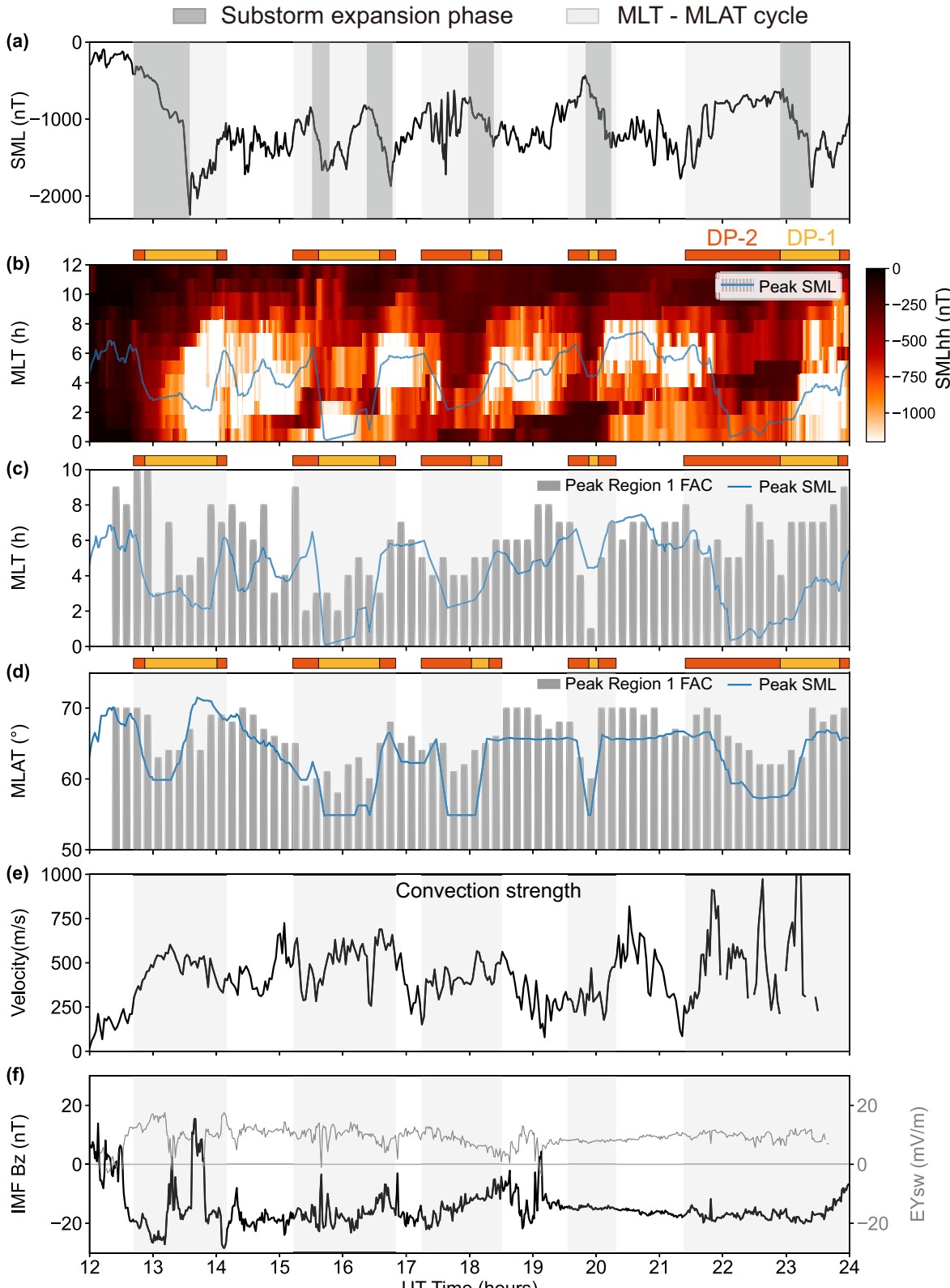

**Fig. 3 | Coordinated evolution of dawnside peak Region 1 FAC and peak SML (westward AEJ) during consecutive storm-time substorms on 17 March 2015.** **a** SML index, with expansion phases marked by dark grey boxes and identified by sharp, monotonic decreases in the SML. **b** Local SML index, with the MLT of the peak SML overplotted. DP-1 and DP-2 intervals for peak SML are identified following[22] as described in the text. The horizontal orange-yellow color bar at the top of the figure indicates the time intervals corresponding to DP-2 and DP-1. **c** Ten-minute-averaged MLT of the peak upward Region 1 FAC and peak SML in

MLT = 0–12. FAC identification is restricted to 50°–70° geomagnetic latitude to exclude high-latitude Region 0 currents. **d** Ten-minute-averaged MLAT of the peak upward Region 1 FAC and peak SML in MLT = 0–12. **e** Mean sunward ionospheric convection speed in the 03-09 MLT sector, derived from SuperDARN global convection maps. **f** IMF $B_z$ and the corresponding $E_y$. Light grey boxe denote the MLT-MLAT cycles of peak Region 1 FAC and peak SML. Source data are provided as a Source Data file.

expanding-contracting motion[48], whereas substorm in our study exhibits clear latitude evolution of currents. Regarding the distinction with pure convection, SMU amplitudes only reach up to roughly half of SML in several events (Supplementary Fig. 1)–suggesting that convection is significant, though SML is not purely a measure of convection.

## Discussion

In this study, we present simultaneous observations of Region 1 FACs, ionospheric convection, and auroral electrojets in the dawn sector during a series of storm-time intense substorms on 17 March 2015. These observations reveal a recurring, large-scale evolution of the current peaks in both MLT and MLAT: an antisunward-equatorward phase followed by a sunward-poleward phase. The longitudinal motion of the current peaks exhibits a mixed nature, combining stepwise and smooth progression: it appears more stepwise in certain intervals (e.g., Figs. 1 and 2), consistent with discrete enhancement and decay of nightside DP-1 currents[44], and smoother during others (e.g., Supplementary Figs. 4 and 5), likely reflecting the continuous evolution of convection-driven DP-2[28,30]. These cycles, summarized schematically in Fig. 4, occur along with enhanced convection and repeat across multiple consecutive substorms. Similar cyclic behavior is also evident in numerous non-storm substorms (Supplementary Figs. 6–8), demonstrating that this cycle of currents is not restricted to storm-time conditions.

The latitudinal evolution of these cycles is consistent with the ECPC framework[39,40]. In this context, the equatorward phase reflects intervals of dominant dayside reconnection, whereas the poleward phase reflects intervals of dominant nightside reconnection. The resolved latitudinal motions of currents occur on closed field lines equatorward of the OCB, not at the OCB itself. Both the Region 1 FAC system[28,29,33,34,53] and the auroral electrojet map to the sunward-convection region on closed field lines (Figs. 1 and 2). Previous ECPC studies also showed longitudinal evolution of convection and currents[27,28,30]. In particular, antisunward propagation of enhanced ionospheric convection following southward IMF turnings has been observed[27,28]. ECPC-based modeling of Region 1 FACs (excluding DP-1-related currents) shows local-time evolution of current maxima as the balance between dayside and nightside reconnection varies[30].

To understand the longitudinal evolution, we classify the AEJ into DP-1 and DP-2 components following[22], a classification fully compatible with the ECPC paradigm[26]. DP-2 represents the convection-driven electrojet and DP-1 corresponds to the nightside unloading current. The antisunward progression of the FAC/SML peak results from two interrelated processes: (1) antisunward expansion of convection-related DP-2 driven by enhanced dayside reconnection[21,27,28], present in all cycles, and (2) subsequent intensification of DP-1 currents[44] within MLT sectors already reached by DP-2, as observed in the first two cycles and consistent with strongly driven substorms[17]. The sunward phase reflects the decay of nightside DP-1[44] combined with the sunward extending of pre-existing currents. The latter can stem from convection-driven currents or from sunward broadening of the DP-1 electrojet[22,54]. During the sunward phase, DP-1 either persists from the preceding interval or newly develops within it, depending on the cycle.

Taken together, these observations establish an explicit connection between the ECPC paradigm and the DP-1/DP-2 paradigm, providing constraints on how DP-1 develops within the global reconnection process. The results indicate that DP-1 onset (expansion onset) can occur during intervals dominated by either dayside or nightside reconnection, while its full evolution commonly involves nightside-dominated reconnection. In this study, auroral electrojet indices are used to identify the expansion phases; future SMILE auroral imaging will help refine the timing of the substorm expansion.

These findings address the global process that organize the full evolution of the expansion phase, a main science objective of the

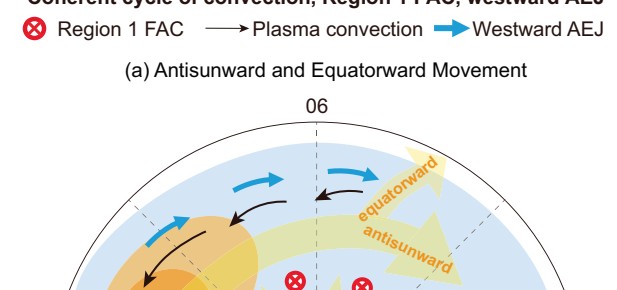

**Coherent cycle of convection, Region 1 FAC, westward AEJ**

⊗ Region 1 FAC ⟶ Plasma convection ➡ Westward AEJ

(a) Antisunward and Equatorward Movement

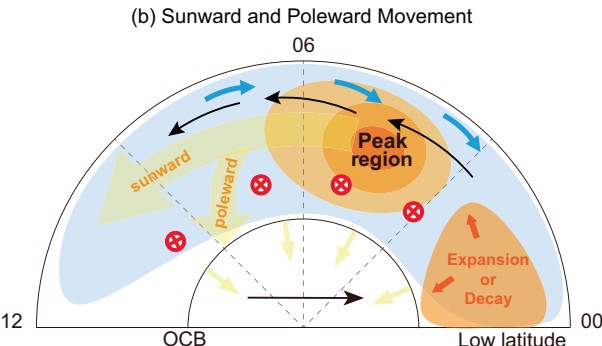

(b) Sunward and Poleward Movement

**Fig. 4 | Schematic illustration of the coherent MLT-MLAT cycle of Region 1 FAC and the westward AEJ, closely coupled to global convection. a** First half of the cycle: the peaks of the FACs and AEJ migrate antisunward and equatorward. This motion can exhibit a mixed character, appearing stepwise due to the discrete addition of nightside DP-1-related currents, or continuous reflecting the continuous evolution of convection-driven DP-2 currents. **b** Second half of the cycle: the peaks of the FACs and AEJ migrate sunward and poleward. Stepwise motion during this phase can arise from the discrete decay of nightside DP-1 currents, whereas smoother motion reflects the overall redistribution of DP-2 currents together with possible spatial expansion of DP-1. The red circled cross denotes the Region 1 FAC, black arrows represent plasma convection, and the blue arrow indicates the westward AEJ. Orange and blue shading indicates the intensity distribution of electric currents, with deep orange marking peak regions and blue denoting weaker, more extended areas. Yellow arrows indicate the directional migration of the parameters in MLT and MLAT.

SMILE mission. The first two substorm expansion occurs largely during an interval of enhanced convection dominated by dayside reconnection, consistent with strong solar-wind control[15–19]. The four subsequent substorm expansions take place mostly during intervals when nightside reconnection dominates, consistent with either enhanced magnetotail reconnection following flux accumulation[8,55], or reduced dayside reconnection associated with northward IMF turnings[13]. In all cases, plasma instabilities and localized convective flows likely shape the DP-1 in the expansion phase as it evolves in the global convection[9,10,12,24,56–62].

## Methods
### Solar wind observations
Solar wind magnetic field and plasma velocity data were obtained from the Time History of Events and Macroscale Interactions during Substorms (THEMIS) B spacecraft, using the Fluxgate Magnetometer (FGM) and Electrostatic Analyzer (ESA), respectively[63]. At the time of the event, THEMIS-B was located approximately 45 Earth radii ($R_E$) upstream of Earth. Data were smoothed to a 1-minute cadence and

time-shifted by 5 minutes to account for solar wind propagation to the dayside magnetopause (assumed at 10 $R_E$), ensuring alignment with the OMNI time axis.

## Geomagnetic Indices and Auroral Electrojets

The geomagnetic storm was characterized using the SYM-H index. Substorm activity was monitored by the SuperMAG SML index[41–43], constructed from an expanded global network of ground-based stations spanning 40°–80° geomagnetic latitude. SML corresponds to the most negative N-component perturbation among all stations at a given time, providing a high-resolution proxy for the peak AEJ intensity. The MLT and MLAT of the station recording the minimum SML value were used to characterize the peak location of westward AEJ. All indices were used at a 1 min cadence.

## Field-Aligned Currents from AMPERE

Field-aligned current (FAC) distributions were obtained from the Active Magnetosphere and Planetary Electrodynamics Response Experiment (AMPERE)[64]. AMPERE derives global FAC maps by combining magnetic perturbation data from about 70 Iridium low-Earth orbit satellites using spherical harmonic fitting. Maps are provided at a 10-minute cadence with a spatial resolution of 1° in MLT and MLAT. To suppress noise signals, current densities below $0.2\,\mu A\ m^{-2}$ were excluded from display.

## Ionospheric convection from SuperDARN

Global ionospheric convection patterns were derived from Super Dual Auroral Radar Network (SuperDARN) observations. Line-of-sight plasma velocity measurements from multiple high-latitude radars were assimilated using statistical fitting techniques to reconstruct 2 min resolution global convection maps[65–67]. To match the cadence of AMPERE data, the convection maps in Figs. 1 and 2 were resampled to 10-minute resolution.

SuperDARN data were processed using the Radar Software Toolkit (RST v5.0), following the standard Map-Potential workflow as in ref. 68. The procedure was as follows. Northern Hemisphere "raw-acf" data from all available radars were first acquired and converted to FITACF format. The FITACF files were processed into combined grid files using a fixed scan length of 60 s to ensure uniform scan segmentation. These grid files were converted into convection map files in the default AACGM-v2 coordinates. Because parts of the OMNI solar wind data were missing, 1-min THEMIS-B solar wind data were used as interplanetary parameters; the THEMIS data were shifted by 5 min to match OMNI timing. Kp index is also included as input. Finally, background statistical model vectors were added, employing the TS18 model[69] with a spherical harmonic order of 8.

## Connection between convection, FACs, and AEJs

Large-scale plasma convection is closely linked to Region 1 FACs. Modeling studies show that Region 1 FACs map not only to the region near OCB, but also to sunward convection on closed field lines deeper inside the magnetosphere[28,29,33,34]. These currents are driven by vorticity or shear in convective flows and can be described as[24,70]:

$$J_{\parallel, i} = B_i \int_{eq}^{i} \frac{\rho}{B} \frac{d}{dt}\left(\frac{\Omega}{B}\right) ds, \qquad (1)$$

where $J_\parallel$ is field-aligned currents, $B$ is magnetic field strength (subscript $i$ indicates the ionosphere), $\Omega$ is flow vorticity, and $\rho$ is mass density. Conceptually, this expression shows that Region 1 FAC enhancements tend to coincide with regions of intensified convection.

The westward AEJ reflects contributions from DP-2 and DP-1 current systems. In the DP-2 regime, the dawnside westward AEJ naturally arises from the sunward return flow of the dawn convection cell[21,71]. Strengthening magnetospheric convection enhances the ionospheric convection electric field and increases large-scale FACs. DP-1 is related to enhanced ionospheric conductivity due to aurora precipitation.

## Data availability

Source data are provided with this paper. OMNI and THEMIS data are available at NASA's Coordinated Data Analysis Web (CDAWeb, https://spdf.gsfc.nasa.gov/pub/data/). The AMPERE field-aligned current is available on https://ampere.jhuapl.edu/browse/. The relevant geomagnetic indices of SuperMAG come from https://supermag.jhuapl.edu/indices/. SuperDARN data can be accessed at https://doi.org/10.20383/102.0447. Source Data are provided with this paper and available at 10.5281/zenodo.18493818.

## Code availability

The radar software toolkit (RST) used to produce SuperDARN convection maps is available at https://doi.org/10.5281/zenodo.7467337, and reference therein. IDL SPEDAS used for analyzing data are freely available at https://themis.igpp.ucla.edu/software.shtml. IDL and Python code used for this study are available at 10.5281/zenodo.17776738.

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

## Acknowledgements

L.D is jointly supported by NSFC grants (42425404) and the National Key R&D Program of China 2025YFF0512100, NSFC grants (42527802,42188101), the Specialized Research Fund for State Key Laboratories of China, and the Strategic Pioneer Program on Space Science II, Chinese Academy of Sciences, grants XDA15350201, XDA15052500. M.H.Z. is supported by NSFC grants 42404178. We acknowledge the use of SuperDARN data. SuperDARN is a network of radars funded by national scientific funding agencies of Australia, Canada, China, France, Italy, Japan, Norway, South Africa, the United Kingdom, and the United States of America. We thank SuperMAG for providing geomagnetic station data and derived geomagnetic indices. We thank the AMPERE team and the AMPERE Science Data Center for providing data products derived from the Iridium Communications constellation. Thanks to the THEMIS mission for providing solar wind data in this event, the Kyoto World Geomagnetic Data Center for SYM-H data.

## Author contributions

L.D. conceptualized the study, analyzed and interpreted data, and wrote the manuscript. T.H.W. analyzed and interpreted data and contributed to manuscript writing. Y.R. collected and processed observational data and contributed to data interpretation. M.H.Z. and J.J.L. processed figure data. W.G., S.W., and C.P.E. contributed to data interpretation. C.W., X.W., and K.L.W. contributed to manuscript revision. All authors reviewed and approved the manuscript.

## Competing interests

The authors declare no competing interests.
