## [Transparent Peer Review file · Nature Communications]

Substorm Expansion Embedded in a Global Cycle of Field-Aligned Currents and Auroral Electrojets

Corresponding Author: Professor Lei Dai

Version 0:

Reviewer comments:

Reviewer #1

(Remarks to the Author)
(General comments)

This paper presents the evolution of field-aligned currents (FACs) observed by AMPERE satellite constellation at low altitudes together with ionospheric convection measured by SuperDARN radar network. The results show that during substorm expansion phases, the peak latitude of the FACs shifts equatorward and toward the nightside, while during the recovery phase, it moves poleward and toward the dayside. In other words, the locations of the FAC and auroral electrojet peaks trace out a full cycle over the course of the expansion and recovery phases. The authors suggest that the substorm should be understood as a process embedded within this global cycle of current system evolution. The concept of large-scale cycle of the peak locations is very interesting and likely new (if it exists). This perspective potentially provides a unifying framework that links the temporal development of substorms to the spatial reconfiguration of the coupled magnetosphere-ionosphere system. However, there are 3 major issues as summarized below. First, the evidence presented does not fully support the existence of the proposed cycle. The issue comes from inadequate color codes of the plots. Second, the authors do not clearly distinguish between a genuine shift of the peak location and the appearance or disappearance of additional FACs. The latter one is well known, and many studies have been accomplished. Third, although the cycle is explained in terms of the motion of the open-closed boundary (OCB), this may not fully account for the observed shifts. A more convincing explanation would be necessary. For these reasons, I believe that the paper requires major revision before it can be considered for publication.

(Specific major comments)

The first concern relates to the supporting evidence for the proposed cycle. In Figure 1, the FACs are shown, but the bluish color scale appears saturated, making it difficult to clearly identify the peak location. I recommend adjusting the color scheme to enhance the visibility of the peaks so that the key feature can be more reliably recognized. In the same figure, the ionospheric plasma velocity is presented, but the auroral electrojet (ionospheric current) is not. Since the ionospheric current depends on ionospheric conductivity, the peak location of the plasma flow does not necessarily coincide with that of the ionospheric current. Furthermore, in Figures S2, S3, and S4, the SML index is shown as a function of MLT and time. These plots do not appear to demonstrate the proposed cycle in the peak MLT, and therefore do not provide clear supporting evidence for its existence.

The second concern involves the nature of the shift in peak locations. During a substorm, multiple types of FACs are expected to emerge. In particular, during the expansion phase, additional strong FACs are known to appear on the nightside. The use of the term "shift" suggests that a specific type of FAC moves to a different location. However, an alternative explanation is that additional FACs appear, thereby producing an apparent "change" in the peak locations. Gjerloev et al. (2004, <https://doi.org/10.5194/angeo-22-2135-2004>) demonstrated that the MLT of the peak auroral electrojet is located near dawn before expansion onset, but is changed on the nightside as the expansion phase progresses, most likely due to the emergence of additional FACs associated with the substorm. It is therefore reasonable to consider that, at the beginning of the expansion phase, additional FACs develop on the nightside, leading to a shift of the peak location in that direction. Conversely, when the recovery phase begins, these additional FACs decay or disappear, producing a shift of the peak location back toward the dayside. I recommend comparing the present results with those of Gjerloev et al. (2004) and discussing both the similarities and the differences.

The third concern relates to the explanation of the proposed cycle. The authors attribute the expansion and contraction of the

peak locations to the motion of the open–closed boundary (OCB). While the OCB framework may account for the latitudinal shift of the peaks, it does not adequately explain the longitudinal shift. How can the observed longitudinal variation be interpreted within this model? In addition, the changes in the magnitudes of the FACs and plasma flows are not sufficiently addressed. A more convincing explanation is needed to clarify these aspects of the cycle. In particular, considering the role of additional FACs could provide a more reasonable interpretation of both the latitudinal and longitudinal "shifts." In addition, within the OCB framework, Region 1 FACs are generally expected to map directly onto the OCB. However, in Figure 3, the Region 1 FACs are shown equatorward of the OCB. Is there a specific physical reason for this apparent separation between the OCB and the Region 1 FACs?

(Minor comments)

Title:

Title does not adequately reflect the content of the study. Since the proposed cycle explicitly includes both the expansion and recovery phases, the title should be revised to make this clear. In addition, the meaning of the phrase "planetary-scale auroral current" is unclear. Does "auroral current" refer specifically to the auroral electrojet, or to the global ionospheric current system more broadly? It is already well established that substorm expansion occurs in regions where ionospheric currents flow, so the present wording of the title is confusing and potentially misleading. A clearer and more precise formulation is needed to properly reflect the scope of the study.

Page 2, "Despite decades of study, the global physical mechanisms governing this cycle—particularly the expansion phase—remain unresolved.":

What specific aspects of the expansion phase remain unresolved? Moreover, what does this paper contribute toward resolving those points?

In introduction: DP1 and DP2 are introduced in the introduction, but these terms are not used for the explanation. To improve consistency and clarity, I recommend revising the introduction so that these terms are meaningfully connected to the analysis and discussion presented later in the paper.

Page 3, The term "conjugate" is used, but its meaning is unclear in this context. Do the authors mean simultaneous observations in the same region, or something else? This should be clarified to avoid ambiguity.

Page 3, "Global MHD simulations provide time-resolved predictions [42, 43], but need observational validation.":

The statement is too general. I recommend specifying which aspects of the simulations require validation, for example, the location of FACs, the dynamics of the ionospheric convection, or the temporal evolution of the substorm phases. Providing concrete points will make the motivation clearer and strengthen the connection between the simulations and the observations presented in this study.

Page 3, "These cycles involve coherent shifts of current peaks across magnetic local time and latitude—initially toward the nightside and equator, then reversing toward the dayside and pole.":

This statement appears inconsistent with the data presented. For the 17 March 2015 event, the current peaks are not shown.

Page 4, "solar wind IMF" reads as "IMF".

Page 4, I recommend rephrasing "nightsideward".

Page 5, " Once nightside reconnection dominates, the OCB—traced by the poleward edges of Region 1 FACs and convection—retreats poleward. The peak of convection and its associated FACs subsequently propagate toward the dayside, reflecting a transition to tail reconnection–dominated convection.":

I recommend revising this sentence using more physically sound terminology rather than imaginative wording. As written, the meaning is unclear, and I am not able to fully understand the intended physical interpretation. A clearer, physics-based explanation is necessary.

Page 5, " Five distinct cycles are highlighted by blue boxes.":

What specific criterion was used to define and select these cycles? The authors should clarify the methodology or threshold employed

Page 6, "all five substorm expansions occur within these cycles.":

For the second one the tendency is opposite. During the expansion phase, the peak moves to later MLTs, and during the recovery phase, it moves to earlier MLTs.

Page 6, "This consistent relationship suggests that substorm expansions are modulated by the broader cyclic reconfiguration of auroral current systems.":

The statement seems to lack sufficient basis. There is a significant leap from the results presented to this conclusion. The authors should provide a clearer justification or additional evidence to support this interpretation, or else rephrase the statement in a more cautious manner. The authors should also consider the possibility that the substorm activities results in the cycle of the peak locations, as Gjerloev et al. (2004) show.

Page 6, Figure S2, S3 and S4:

The auroral electrojets dominant, and Figure S2, S3 and S4 seem not to show such cycle clearly.

Page 6, " Our central result is that substorm expansion is embedded within a planetary-scale cycle of auroral current systems that are tightly coupled to global plasma convection."

The statement is problematic. Numerous previous studies have already shown that substorm expansion occurs in the region where convection is present. Therefore, the novelty and physical meaning of the claim are unclear. The authors should explicitly define what is meant by "planetary-scale cycle." How does the cycle described here differ from earlier concepts, particularly the framework proposed by Kamide and Kokubun (1970, <https://doi.org/10.1029/96JA00142>)? Observations also show that some expansion onset takes place near the Harang discontinuity, which is located in the ionospheric convection (Weygard et al., 2008, <https://doi.org/10.1029/2007JA012537>). In addition, global MHD simulation results suggest that the substorm expansion is initiated and sustained in the convection system (Tanaka et al., 2010, <https://doi.org/10.1029/2009JA014676>; Ebihara and Tanaka, 2017, <https://doi.org/10.1002/2017JA024294>; Tanaka et al. (2021, <https://doi.org/10.1029/2020JA028942>). I recommend discussing these previous perspectives.

Page 7, " dominant dayside reconnection opens magnetic flux, expanding the polar cap and pushing the OCB equatorward." What do you mean by "dominant"? Do you mean that if the dayside reconnection is minor, the polar cap location is unchanged?

Page 7, "closing magnetic flux and contracting the polar cap, leading to a poleward retreat of OCB, auroral currents and convection. Correspondingly, nightside-driven convection and Region 1 FAC progress toward the dayside." Please explain the reason why contracting polar cap results in poleward retreat of auroral currents and convection? The convection flows entirely at all magnetic latitudes from the magnetic pole to the magnetic equator.

Page 7, "the first substorm expansion occurs within convection driven by dominant dayside reconnection.": Is this statement based on evidence, or speculation?

Page 7, " Such substorm is likely directly coupled to real-time solar wind drivers": I recommend revising this statement. What do you mean by real-time?

Page 7, "the four subsequent substorm expansions take place during convection when nightside reconnection dominates.": Is this statement based on evidence, or speculation?

Page 8, " unified framework in which reconnection-driven global convection modulates substorm expansion, while mesoscale processes shape its onset timing and substructure.": The statement lacks a clear basis. It is already well established that substorm expansion takes place within the large-scale convection system. The authors should clarify what new evidence supports this claim, and specify the major difference between their proposed framework and existing knowledge.

Page 8, "but part of a global reconnection-modulated convection cycle that reconfigures auroral current systems": Same as above.

Page 9, "Region 1 FACs originate from the low-latitude magnetospheric boundary layer [54], and typically map to sunward convection flows on closed field lines [55, 56]."

Do [55, 56] actually explain why FACs originating from the boundary are mapped into the closed region? In those studies, the assumption is that FACs propagate strictly along magnetic field lines, implying that boundary-originating FACs should map to the OCB in the ionosphere, rather than closed regions.

Page 10, Equation (1):

This equation includes time derivative. Is it possible to explain the Region 1 FACs in which the time variation is not so high.

Page 10, "Conceptually, this expression shows that Region 1 FAC enhancements tend to coincide with regions of intensified convection.":

The convection is the circular motion of plasma in the magnetosphere. Do you intend to mean that the Region 1 FACs are generated in the magnetosphere, rather than the boundary layer? [56] considers flow shear around the boundary to cause the Region 1 FACs.

Page 10, "Strengthening magnetospheric convection enhances the ionospheric convection electric field and increases large-scale FACs, which serve as a proxy for auroral precipitation.":

Do you mean that magnetospheric convection, not plasma flows at boundary, enhances Region 1 field-aligned currents (FACs), or the ionospheric convection enhances Region 1 FACs? What is the relation to the southward component of IMF? How is this related to auroral precipitation? As I understand it, the authors are focusing on the dawnside Region 1 downward FACs. In the downward FAC region, significant auroral precipitation is generally not expected.

Page 10, "The AMPERE field-aligned current is available on <http://ampere.jhuapl.edu/rBrowse/index.html>.": This site is inaccessible as of September 1.

Page 10, " SPEDAS codes used for analyzing THEMIS data are freely available at <http://spedas.org/blog/>": The site is also inaccessible as of September 1.

Reviewer #2

(Remarks to the Author)

Review of "Substorm Expansion Embedded in a Planetary-Scale Auroral Current Cycle"

This paper presents observations of auroral current systems and ionospheric convection, alongside indicators of magnetospheric substorm activity. The auroral current systems and convection data provide supporting evidence for the expanding contracting polar cap (ECPC) paradigm. In brief, the ECPC states that intervals of dominant low-latitude dayside reconnection enhance currents and flows on the dayside, leading to polar cap expansion, and intervals of dominant nightside reconnection enhance currents and flows on the nightside and lead to polar cap contraction. Low-latitude dayside reconnection is driven by coupling with a southward IMF, and nightside reconnection is commonly expected during substorms. Which of these is dominant is typically dependent on the magnitude of negative IMF Bz and the magnitude of any concurrent substorm activity.

The first event identified in this study has a substorm occurring during strongly southward IMF. As such, dayside reconnection is observed to be dominant despite the occurrence of the substorm. In subsequent substorms, nightside reconnection is observed to be dominant. It is claimed that these results "uncover a reconnection-driven convection process that modulates substorm expansion on planetary scales". This seems to be a rather inflated claim. The ECPC paradigm fully predicts the reported behaviour. Whilst the observations presented here are consistent with expectations, they do not uncover anything unexpected.

It is not clear what the "planetary" scale modulation of substorms is. There is not really any evidence provided that the reconnection cycle is "modulating" substorm expansion. It is expected by the ECPC paradigm that substorms will modulate the currents and convection, as shown here (this has of course been well-reported).

Perhaps the substorm expansion itself could be modulated by the concurrent upstream conditions and resulting magnetospheric dynamics. The MLT and MLAT of each substorm is different, for instance, and further analysis could determine whether those differences had any relation to the ongoing large-scale dynamics. But this would not necessarily be a new result – previous work (e.g. Milan et al., 2009, doi:10.5194/angeo-27-659-2009; Milan et al., 2010, doi:10.1029/2010JA015663) has looked at how the substorm expansion phase evolves under different solar wind driving conditions. That later substorms occur during the ensuing main phase of the geomagnetic storm, with its associated ring current enhancement, is also not considered, but has also been shown previously to influence the evolution of the substorm (Milan et al., 2008, doi:10.1029/2008JA013340).

Overall, although the observations provide a nice example of a well-established phenomenon, there is insufficient detail in the analysis or discussion to establish what genuine advancement to the field these observations might offer.

Minor comments:

On page 4 (section 2.1) it is stated that "Figure 1 presents the spatial and temporal development of Region 1 FACs (blue) and sunward ionospheric flows in the dawn sector." Figure 1 presents a global view of the FACs and convection. The FACs are not only shown in blue but also red. The full figure should be introduced before then focussing on a particular aspect.

In reference to Figure 2: It would be useful to see the IMF Bz component throughout, to assess whether the difference between the "descending" nature of the first substorm and the "ascending" nature of the subsequent ones was related to the concurrent solar wind conditions. Although Bz is shown for a wider interval in the SI, it is not clear. In fact, it might be more useful to present in Fig. 2 the solar wind electric field, which is expected to more directly correlate with the strength of dayside reconnection.

Section 4.3: What statistical fitting technique was used to assimilate the SuperDARN data? A commonly used fitting technique is that described by Ruohoniemi and Baker (1998, doi:10.1029/98JA01288) but this is not referenced. If this is the method used, then further details of the fitting process (radar data gridding options, IMF input, background model, HMB identification etc.), need to be provided for reproducibility.

Reviewer #3

(Remarks to the Author)

Key Results: This manuscript explores the relationship between substorm expansion phases and large-scale plasma convection cycles. Sharp decreases in the AL index identify the substorm expansion phases, while convection cycles are determined using AMPERE-derived pictures of the field-aligned currents (FACs), as well as Super-DARN-derived convection maps. The authors conduct an event study using five substorms that occurred during the 17 March geomagnetic

storm. Initially, they investigate the first of the five substorms. They demonstrate that during the expansion phase of this substorm, the region-1 (R1) FACs and convection signatures migrate equatorward (Fig. 1). At the same time, their peak intensities shift nightward from the pre-noon sector to the post-midnight sector. This is interpreted, following the expanding/contracting polar cap (ECPC) model, as an equatorward expansion of the open-closed boundary (OCB), implying that dayside/magnetopause reconnection is the dominant driver of magnetospheric convection during this interval. Thus, the resulting convection proceeds from the dayside towards the nightside. About ten minutes later, during the late expansion and recovery phases of the substorm, the inverse behavior is observed, with the FACs and convective flows moving poleward and shifting back toward the dayside, implying a contracting OCB, with nightside/tail reconnection driving the convection patterns.

The next section of the manuscript builds upon this analysis and extends it to all five substorms. To further illustrate the pattern of convection cycles, the authors plot the time series of the location of the peak intensities of the R1 FACs and the SML index (a metric for the strength of the westward auroral electrojet commonly attributed as the ionospheric portion of the substorm current wedge) along with the convection strength (Fig. 2). They demonstrate that each substorm expansion is embedded within a longer (~1–2 hours) convection cycle. The peaks in the FACs and westward auroral electrojets consistently shift nightward/sunward and equatorward/poleward during the “descending”/“ascending” phase of these convection cycles. The “descending” phase of the convection cycle equates to the dayside-dominated convection and the “ascending” to nightside-dominated, as is summarized in Fig. 3. Interestingly, the first substorm expansion phase occurs during the “descending” phase of the convection cycle. In contrast, the following four all occur during the “ascending” phase. The authors contend that the first substorm is likely directly driven, as described by Dai et al. (2023), while the remaining substorms result from nightside/tail reconnection.

Validity: My critique of this study hinges on the fact that the five substorms considered occur during a strong geomagnetic storm, indeed, the strongest storm of Solar Cycle 24. As strong storms are generated by abnormally strong geomagnetic driving, the results of this study may not be generalizable to all traditional and non-storm-time substorms.

Akasofu’s (1964) original description of substorm onset included a sudden brightening of the pre-onset auroral arc, followed by a localized auroral bulge forming near midnight local time that expands poleward and azimuthally as the expansion phase progresses. Although it is commonplace in the community to now identify substorm onsets using sudden decreases in the AL or SML indices, as done in this study, this approach may identify substorms that differ in character from the traditional auroral picture, especially during storms. The strong and often long-duration solar wind driving muddies the situation, making it difficult to differentiate between the various substorm phases and other substorm-related phenomena such as pseudobreakups and steady magnetospheric convection (SMC) intervals. For example, the second shaded segment in Fig. 1, described as the recovery phase of the first substorm, could be interpreted as an SMC interval. Walach & Milan (2015) argue that most SMC intervals are simply substorms that continue to be driven throughout what would otherwise be their expansion and recovery phases. This has led some researchers to introduce a new classification termed “multiple intensifications” (Milan et al., 2021, 2024). Using this new classification, Bower et al. (2025) expanded upon the Substorm Onsets and Phases from Indices of the Electrojet (SOPHIE) algorithm (Forsyth et al., 2015) to include multiple intensifications. Termed SOPHIE-M, this algorithm labeled all five substorms identified in this study as multiple intensifications. Further complicating our understanding of storm-time substorms, Zou et al. (2025) recently demonstrated that very intense storm-time substorms may not be traditional substorms at all, but rather global-scale convection events distinct from traditional substorms.

These critiques do not negate the primary results and interpretation of the study. Substorms, SMCs, and multiple intensifications are not inherently distinct modes. Instead, in my opinion, they represent a continuum of substorm-like processes in the magnetosphere. The distinction between when a recovery phase is called an SMC and when numerous expansion phases are labeled as multiple intensifications is arbitrary. However, more care should be taken to avoid generalizing the results to all traditional and non-storm-time substorms. For example, in the abstract, the sentence “Based on coordinate observations of a series of intense substorms...” should be “Based on coordinate observations of a series of intense storm-time substorms...”. Furthermore, the Discussion section should outline some of the study’s limitations, for example, in relying on auroral electrojet indices to determine substorm expansion rather than using auroral imaging. The authors could also expand upon the description of the three other intervals included in the Supplementary Information. Currently, they are briefly mentioned in a single sentence. Fig. S2 and S3 are both during non-storm intervals, and many of the highlighted substorms appear to be simpler and more akin to the traditional substorm description based on the AL profiles. Simply stating that the Supplementary Information contains two intervals of non-storm-time substorms would justify the generalizability of the main text’s findings.

Significance: Substorm expansion phases are among the most significant magnetospheric space weather events, resulting in an explosive global-scale reconfiguration of the magnetosphere on timescales of tens of minutes. Their linkage to the ionosphere produces intense aurorae, ionospheric currents, and perturbations in the ground magnetic field. As such, understanding the physical mechanisms driving substorms is critical to eventual space weather modeling and prediction. As the authors state in the introduction, much of substorm research has focused on exploring substorm onset mechanisms and the micro-scale details of reconnection. Meanwhile, relatively little attention has been devoted to understanding the global-scale mechanisms and effects of substorms, particularly during the expansion phase. This is mainly because in-situ spacecraft observations, such as those from MMS and THEMIS, only provide a micro-scale view of the global magnetosphere-ionosphere (MI) system. However, global-scale datasets do exist but have been underutilized by the scientific community; for example, the AMPERE-derived pictures of the FACs, collections of ground-based magnetometers capable of resolving the impact of ionospheric currents, which are distilled into geomagnetic indices, and SuperDARN-derived ionospheric convection maps. Therefore, studies like this address a gap in our scientific understanding of substorms

and help contextualize micro-scale investigations.

Data and Methodology: The data and methodology are sound. All the datasets used in this study are widely employed by the scientific community and readily accessible. The methodology used is straightforward, easy to understand, and is described in sufficient detail to support replication of the results. The proper links to the data and source code are also provided.

Analytical Approach: The study relies on coordinated data observations and does not employ statistical or other analytical approaches. As such, this section of the review does not apply to this manuscript.

Suggested Improvements:

My major suggestions have been discussed in the Validity section above. Some minor suggestions are:

Fig. 1: It would be helpful to add a horizontal line in panel (a) at $B_z = 0$ to distinguish when the IMF is southward vs. northward.

Fig. 2: To understand the context of convection cycles and substorm expansions in relation to solar wind driving, it would be helpful to add a panel showing the IMF B_z , similar to Fig. 1a. If this makes the figure too large, panels (b) and (c), as well as panels (d) and (e), could be combined by overplotting the yellow lines overtop of the bars. This may also make the correlations between these sets of panels more evident.

Fig. S2–S4: To understand the storm context of these substorm expansions and convection cycles, it would be helpful to add additional panels showing the Sym-H index similar to Fig. S1c.

Clarity and Context: The manuscript is well-written, clear, and develops in a logical and readily understandable fashion. The introduction is concise yet nicely frames the study within its historical and scientific context.

References: The manuscript provides ample references, several of which I consulted during the evaluation of this manuscript.

Your Expertise: The reviewer is an expert in empirical magnetic field modeling of planetary magnetospheres. In particular, the reviewer has used flexible empirical magnetic field models in conjunction with data mining approaches to resolve global-scale magnetospheric dynamics during storms and substorms.

Version 1:

Reviewer comments:

Reviewer #1

(Remarks to the Author)

[Major comments]

My first major concern in the previous round was the supporting evidence for the proposed cycle. I acknowledge that Figure 1 has been substantially improved, and it is now possible to identify the evolution of the FACs more clearly. The inclusion of equivalent current maps is also helpful. I agree that the Region 1 FACs and the westward auroral electrojet exhibit a coherent antisunward and equatorward progression during the expansion phase. However, an alternative interpretation remains possible: The Region 1 FACs and the auroral electrojet may develop discretely rather than continuously. In Figures 1c and 1d, the dayside FACs remain strong (as highlighted by the green circle), while enhanced nightside FACs appear separately (as indicated by the grey arrow). This behavior suggests the formation of distinct current systems, rather than a smooth, continuous equatorward and antisunward migration. Therefore, I am not fully convinced that the FAC and auroral electrojet peaks move equatorward and antisunward in a smooth manner, as schematically depicted in Figure 3. Additional clarification or quantitative evidence would be necessary to distinguish between continuous motion and the discrete development of multiple current systems. Figure 2b, which shows the MLT variation of the auroral electrojet, also exhibits discrete changes, rather than a smooth evolution in MLT. If the evolution of the FACs and the auroral electrojets is indeed discrete, this interpretation would be consistent with the authors' revised description (Lines 169–175). In that case, however, the novelty of the proposed "cycle" needs to be clarified more explicitly, particularly in relation to previously reported discrete developments of substorm-related current systems. If a smooth migration cannot be clearly demonstrated, the applicability of the ECPC framework would become unclear, and a more careful and internally consistent physical explanation would be required.

[Minor comments]

I raised a concern about the use of the term "planetary-scale auroral current." In their response, the authors state: "In this study, the phrase 'planetary-scale' denotes the global extent of the observed current cycle across a wide range of magnetic local times. The phrase "auroral currents" denotes the combined system of auroral electrojets within the ionosphere and the auroral field-aligned currents (FACs) that connect the ionosphere to the magnetosphere. Auroral electrojets are widely considered a component of the auroral current system, and the term 'auroral FACs' has also been used in previous studies (e.g., Lysak, 1985; Elphic et al., 1998; Juusola et al., 2016; Xiong et al., 2021). To avoid ambiguity, we have clarified these definitions in the Introduction." The term is not well clarified in the Introduction section. Aurora refers to an optical

phenomenon, whereas current is a physical quantity describing the flow of electric charge per unit time. The physical connection between aurora and current is therefore not self-evident in the present paper and requires careful clarification. While some previous studies (e.g., Lysak and Elphic) have used the term "auroral FACs," it is clear in those works that the intention was to describe FACs associated with electron precipitation. In contrast, the present paper focuses primarily on downward FACs, in which the relation between FACs and field-aligned currents are unclear. This distinction is probably important and is not sufficiently addressed. Moreover, it is well known that both FACs and the ionospheric equivalent currents (e.g., DP2) can exhibit global-scale structures. In this context, introducing the additional qualifier "planetary-scale" does not seem necessary, nor does it add clear physical meaning. In fact, the term "planetary-scale" may be misleading, as it could be interpreted as implying processes confined near the planet itself, rather than processes involving the magnetosphere as a whole. For these reasons, I again recommend rephrasing or avoiding the term "planetary-scale auroral current", and instead using terminology that more precisely reflects the physical quantities and processes actually analyzed in this study.

Line 116-118: "Global simulations show that convection-driven Region 1 FACs develop from the dayside toward the nightside [28, 32–34],...":

I recommend rephrasing "convection-driven" by "convection-associated" because [28, 32-34] seems not to state explicitly that the convection drives the Region 1 FACs.

(Remarks on code availability)

Reviewer #2

(Remarks to the Author)

The authors have thoroughly addressed most of the comments raised in the first set of reviews. It is much improved, but on one point I am still not satisfied the authors have fully considered the implications of the existing literature.

In their response to my review they state "we identify a coherent longitudinal (MLT) cyclic evolution of auroral currents. To our knowledge, such MLT evolution has not been described or discussed in the ECPC literature" and "we show that the auroral current system exhibits a longitudinal (MLT) cyclic evolution, adding spatio-temporal dynamics that ECPC does not address".

In the revised manuscript they write: "Notably, the cycles display a coherent longitudinal (MLT) movement of the current peaks—an antisunward motion during the equatorward phase and a sunward return during the poleward phase. This longitudinal evolution of currents has not been reported and discussed in previous ECPC studies".

It is implicit in the ECPC paradigm that the foci of the convection cells (and hence the peaks of the FAC) evolve in local time as the dayside (solar wind driven) and nightside (substorm driven) reconnection dominance varies. When magnetopause reconnection is dominant the convection enhancement, that begins on the dayside, then expands antisunward, whilst the polar cap also expands. When nightside reconnection is dominant the convection enhancement, that begins on the nightside, then expands sunwards, whilst the polar cap contracts. This is exactly what is described by the authors, above.

Studies of a local time evolution of the FAC (or convection cells) are not new. Lockwood et al. (1986) provided some of the earliest evidence of flows which "after a southward turning of the IMF were propagated... around the afternoon sector". More recently, Milan (2018) modelled the evolution of the FAC through cycles of the ECPC paradigm. His model "can be used gainfully to understand the factors that determine region 1 and 2 current intensities and, most importantly, the dynamics of the current systems under different solar wind-magnetosphere coupling conditions. These include changes in the latitude of the current regions as the polar caps expand and contract and the local time of the current maxima as dayside and nightside reconnection rates vary."

As such, I think it would be appropriate to acknowledge how the observations presented in this paper – including an MLT evolution – can in fact be interpreted in terms of the ECPC paradigm. That is not to say that the other interpretations offered by the authors, such as the varying dominance of the DP1 / DP2 systems, should not also be given due consideration. That said, it is probably also important to note that the DP1 / DP2 classification of the ionospheric current systems is not incompatible with the ECPC paradigm either (e.g. Milan et al., 2017).

References:

Milan, S. E. (2013), Modeling Birkeland currents in the expanding/contracting polar cap paradigm, *J. Geophys. Res. Space Physics*, 118, 5532–5542, doi:10.1002/jgra.50393.

Lockwood, M. et al. (1986), Eastward propagation of a plasma convection enhancement following a southward turning of the interplanetary magnetic field, *Geophys. Res. Lett.*, 13, 72-75.

Milan, S. E. et al. (2017), Overview of Solar Wind–Magnetosphere–Ionosphere–Atmosphere Coupling and the Generation of Magnetospheric Currents, *Space Sci. Rev.* 206:547–573 doi:10.1007/s11214-017-0333-0.

(Remarks on code availability)

Reviewer #3

(Remarks to the Author)

The authors have made considerable efforts to adequately address all my major and minor comments from the previous review and, in doing so, have significantly improved the clarity of the manuscript. Likewise, the authors have made significant changes to address the other reviewer's comments. As such, I recommend that the manuscript be accepted for publication in its current form.

(Remarks on code availability)

The source code is provided in a Zenodo archive, along with a brief README file that describes the source code and datasets. Overall, the analysis in this study is relatively simple, relying on coordinated data observations, as reflected in the code's simplicity. The majority of the code is dedicated to generating figures from standard, widely used datasets. Although I did not attempt to run any of the code myself, replicating the figures from the code seems straightforward.

Version 2:

Reviewer comments:

Reviewer #1

(Remarks to the Author)

The manuscript has been substantially revised, and the authors have addressed my comments and concerns appropriately. I have no further comments and recommend the manuscript for publication in Nature Communications.

Reviewer #2

(Remarks to the Author)

The authors have now fully addressed all of my comments and concerns outlined in previous reviews.

We sincerely appreciate your feedback, which has been invaluable in guiding our revision. In response, we have expanded data analysis, clarified key points, and refined our interpretations. Detailed, point-by-point responses are provided below.

REVIEWER COMMENTS

Reviewer #1 (Remarks to the Author):

(General comments)

This paper presents the evolution of field-aligned currents (FACs) observed by AMPERE satellite constellation at low altitudes together with ionospheric convection measured by SuperDARN radar network. The results show that during substorm expansion phases, the peak latitude of the FACs shifts equatorward and toward the nightside, while during the recovery phase, it moves poleward and toward the dayside. In other words, the locations of the FAC and auroral electrojet peaks trace out a full cycle over the course of the expansion and recovery phases. The authors suggest that the substorm should be understood as a process embedded within this global cycle of current system evolution. The concept of large-scale cycle of the peak locations is very interesting and likely new (if it exists). This perspective potentially provides a unifying framework that links the temporal development of substorms to the spatial reconfiguration of the coupled magnetosphere-ionosphere system. However, there are 3 major issues as summarized below. First, the evidence presented does not fully support the existence of the proposed cycle. The issue comes from inadequate color codes of the plots. Second, the authors do not clearly distinguish between a genuine shift of the peak location and the appearance or disappearance of additional FACs. The latter one is well known, and many studies have been accomplished. Third, although the cycle is explained in terms of the motion of the open-closed boundary (OCB), this may not fully account for the observed shifts. A more convincing explanation would be necessary. For these reasons, I believe that the paper requires major revision before it can be considered for publication.

(Specific major comments)

The first concern relates to the supporting evidence for the proposed cycle. In Figure 1, the FACs are shown, but the bluish color scale appears saturated, making it difficult to clearly identify the peak location. I recommend adjusting the color scheme to enhance the visibility of the peaks so that the key feature can be more reliably recognized. In the same figure, the ionospheric plasma velocity is presented, but the auroral electrojet (ionospheric current) is not. Since the ionospheric current depends on ionospheric conductivity, the peak location of the plasma flow does not necessarily coincide with that of the ionospheric current. Furthermore, in Figures S2, S3, and S4, the SML index is shown as a function of MLT and time. These plots do not appear to demonstrate the proposed cycle in the peak MLT, and therefore do not provide clear supporting evidence for its existence.

Thank you for pointing out these issues. We have made revisions to the figures to address them directly.

We adjusted the FAC color scale to ensure that the peak amplitude and its motion are clearly distinguishable across MLT. In the updated Figure 1, the FAC maxima are now presented without saturation. Following your suggestion, we have also included a 2D map of local SML (rotated Bn perturbations) in Figure 1. The movement of the SML peak now clearly demonstrates the longitudinal cycle. The cycle of currents in the second substorm is provided in updated Fig.S2.

The difficulty in Figures S2–S4 arose because the pre-expansion SML values were weak and therefore appeared mostly dark in the original color scale, making the cyclic motion difficult to observe. To address this, we added a dedicated panel that displays only the instantaneous peak SML location as a function of time, with all other MLT sectors masked out. This representation makes its longitudinal progression unambiguous. In the revised Figure 2, we also introduced a new panel (panel b) showing an MLT–time map of the SML index overlaid with the tracked peak location. This panel now clearly reveals the longitudinal cyclic evolution in the text.

These revisions clarify the evidence for the cyclic evolution of auroral currents.

The second concern involves the nature of the shift in peak locations. During a substorm, multiple types of FACs are expected to emerge. In particular, during the expansion phase, additional strong FACs are known to appear on the nightside. The use of the term “shift” suggests that a specific type of FAC moves to a different location. However, an alternative explanation is that additional FACs appear, thereby producing an apparent “change” in the peak locations. Gjerloev et al. (2004, <https://doi.org/10.5194/angeo-22-2135-2004>) demonstrated that the MLT of the peak auroral electrojet is located near dawn before expansion onset, but is changed on the nightside as the expansion phase progresses, most likely due to the emergence of additional FACs associated with the substorm. It is therefore reasonable to consider that, at the beginning of the expansion phase, additional FACs develop on the nightside, leading to a shift of the peak location in that direction. Conversely, when the recovery phase begins, these additional FACs decay or disappear, producing a shift of the peak location back toward the dayside. I recommend comparing the present results with those of Gjerloev et al. (2004) and discussing both the similarities and the differences.

We appreciate this important clarification. We agree that substorm expansion can involve multiple types of FACs. Following your recommendation, we have compared our events with the behavior reported by Gjerloev et al. (2004), and the revised manuscript now discusses both similarities and distinctions.

The antisunward motion of the peak FAC is contributed from two processes: the intensification of additional (DP-1-related) nightside FACs as noticed in Gjerloev et al.

(2004), and an additional progressive extension of pre-existing FAC current from the dayside toward the nightside (Fig.1c-d, Fig.S2c-d). Notably, the newly formed nightside FACs tend to strengthen at the MLT sectors where the pre-existing FACs have already reached.

The return of the peak toward the dayside similarly reflects two processes: the decay of the nightside DP-1 FACs as noted in Gjerloev et al. (2004), and also the sunward extension of the pre-existing FAC (Fig.1e-f, Fig.S2e-f). The combined effect produced the observed cyclic MLT motion.

These points have been incorporated into the revised manuscript, in Line 169-176,186-190, 196-205, 213-215, 382-397. The longitudinal evolution is interpreted within the DP-1/DP-2 substorm current framework (Kamide and Kokubun, 1996; Akasofu, 2017), as further elaborated in our response to comment #3.

The third concern relates to the explanation of the proposed cycle. The authors attribute the expansion and contraction of the peak locations to the motion of the open-closed boundary (OCB). While the OCB framework may account for the latitudinal shift of the peaks, it does not adequately explain the longitudinal shift. How can the observed longitudinal variation be interpreted within this model? In addition, the changes in the magnitudes of the FACs and plasma flows are not sufficiently addressed. A more convincing explanation is needed to clarify these aspects of the cycle. In particular, considering the role of additional FACs could provide a more reasonable interpretation of both the latitudinal and longitudinal "shifts." In addition, within the OCB framework, Region 1 FACs are generally expected to map directly onto the OCB. However, in Figure 3, the Region 1 FACs are shown equatorward of the OCB. Is there a specific physical reason for this apparent separation between the OCB and the Region 1 FACs?

We thank you for this important point. We agree that latitudinal OCB expansion/contraction alone cannot account for the observed longitudinal motion of the FAC and SML peaks. In the revision, we have substantially expanded the analysis and discussions to clarify the interpretation of the longitudinal evolution.

First, we identify DP-2 and DP-1 components in Figure 2 using their spatiotemporal signatures, following Kamide & Kokubun 1996. The identification method is illustrated in the figure caption of enclosed Fig. R1 and added in the revised text, line 250-263.

Figure 13. Sketch of variations of the two components of the substorm electrojets. The A component represents the so-called wedge current system, and the B component represents the convection electrojet. The A and B current systems express presumably the unloading process and the directly driven process, respectively. The expansion onset, which signals a sudden increase in the A component, is designated by $T = 0$. Three different curves are shown for A, having three different timescales.

Figure 11. Schematic diagram of the auroral electrojets showing the different roles of ionospheric electric fields and conductivities in the eastward and westward electrojets at different latitudes and local times.

Figure R1. Figure 13 (left) and Figure 11 (right) in Kamide and Kokubun, 1996 are used to identify DP 1 and DP 2 aurora electrojet. In the time domain, DP-1 dominates during the expansion and early recovery phases. In the space domain, the DP-1 electrojet develops on the nightside and extends both eastward and westward, typically spanning MLT=22–04 (Fig.11 in Kamide and Kokubun, 1996, and also Fu et al.,2021JGR). In contrast, the DP-2 electrojet is a global convection-driven system that persists throughout all substorm phases but becomes overshadowed when DP-1 intensifies. Based on the above description, in this study, a SML peak is classified as DP-1 if (1) it occurs during the expansion or early recovery phase, and (2) its MLT falls within the typical DP-1 sector (22–04MLT), which can broaden during strong substorms (Kamide1996JGR, Fu2021JGR). SML Peaks that do not satisfy these DP-1 conditions but coincide with enhanced ionospheric convection ($>300\text{m/s}$) are classified as DP-2.

Second, the antisunward progression of the FAC/SML peak is produced by two combined effects: (1) antisunward extension of the convection-related DP-2 driven by dayside reconnection (e.g. Nishida, 1968, Dai et al.,2024) and (2) the newly formed nightside DP 1/FACs (Gjerloev et al. 2004, Kamide and Kokubun, 1996). This antisunward motion occurs concurrently with the equatorward displacement of the peaks, consistent with a dayside-reconnection–dominated interval within the OCB framework (Cowley and Lockwood 1992; Milan and Grocott,2021).

Third, the sunward phase reflects the decay of nightside DP-1 (Gjerloev et al.,2004) combined with the sunward extending of pre-existing currents. The sunward extending can stem from convection-related currents or from sunward broadening of the DP-1 electrojet (Kamide1996JGR,Milan2010). This sunward motion coincides with poleward movement of the current peaks, indicating a nightside-reconnection–dominated interval (Cowley and Lockwood 1992; Milan and Grocott 2021).

Compared with the earlier version, the revised interpretation now includes the

temporal evolution (growth, expansion, and decay) of the DP-1 system in addition to reconnection-driven convection effects. Modifications are in the discussion section.

Regarding the magnitude of plasma convection, it generally intensifies throughout the cycle. Theoretically, FAC strength depends on the time derivative of the plasma flow shear (Sato and Iijima 1979; Keiling et al. 2009). Lines 513-526. FAC may have a time-delayed response to the evolving convection strength. This study focuses on the cyclic behavior, the aspects of this coupling in the magnitude are left for future work.

Finally, the apparent separation between the OCB and Region 1 FACs in Figure 3 is consistent with previous model (Sonnerup,1980) and our observations. In this model, the Region 1 FACs can originate from the magnetosphere sunward convection in the closed-field-line region (Sonnerup,1980). The mapping of Region 1 FAC to closed field line is also consistent with statistical observations (Wing et al.,2010JGR) and global MHD simulations (Dai et al.,2024; Zhu et al.,2024,2025JGR). In our observations, the Region 1 FAC maxima are often located at 60–70° MLAT—coinciding with the sunward return-flow region in the closed-field line (green circle in Fig. 1)—and thus equatorward of the OCB. Modifications are in line 364-369.

These revisions have been incorporated into the Discussion section. The DP-1–DP-2 analyses have been added to Figure 2.

(Minor comments)

Title:

Title does not adequately reflect the content of the study. Since the proposed cycle explicitly includes both the expansion and recovery phases, the title should be revised to make this clear. In addition, the meaning of the phrase "planetary-scale auroral current" is unclear. Does "auroral current" refer specifically to the auroral electrojet, or to the global ionospheric current system more broadly? It is already well established that substorm expansion occurs in regions where ionospheric currents flow, so the present wording of the title is confusing and potentially misleading. A clearer and more precise formulation is needed to properly reflect the scope of the study.

Thank you for this valuable comment. In this study, the phrase "planetary-scale" denotes the global extent of the observed current cycle across a wide range of magnetic local times. The phrase "auroral currents" denotes the combined system of auroral electrojets within the ionosphere and the auroral field-aligned currents (FACs) that connect the ionosphere to the magnetosphere. Auroral electrojets are widely considered a component of the auroral current system, and the term "auroral FACs" has also been used in previous studies (e.g., Lysak, 1985; Elphic et al., 1998; Juusola et al., 2016; Xiong et al., 2021). To avoid ambiguity, we have clarified these definitions in the Introduction.

In the five substorms observed, the proposed cycle includes parts of the growth or

recovery phases, but does not cover either phase in full. The expansion phase is temporally embedded within the cycle across all events. For this reason, our analysis focuses on the expansion phase. To reflect this emphasis, and to clarify that the cycle itself is planetary-scale, we have revised the title to ‘Substorm Expansion Temporally Embedded in a Planetary-Scale Cycle of Auroral Currents’

Page 2, "Despite decades of study, the global physical mechanisms governing this cycle—particularly the expansion phase—remain unresolved.":

What specific aspects of the expansion phase remain unresolved? Moreover, what does this paper contribute toward resolving those points?

While previous research has focused primarily on identifying onset mechanisms that initiate substorm expansion, much less is known about the global processes that organize the full evolution of the expansion phase. Our study addresses this by identifying a cyclic evolution of auroral currents across both magnetic local time and latitude. Together, the observations suggest that substorm expansion (DP-1) can begin during either dayside-dominated or nightside-dominated reconnection, whereas the full temporal development of DP-1 commonly involves dominant nightside reconnection.

We have revised the text to clearly articulate these points in the Introduction and Discussion sections. Modifications in Line 87-90, 397-401,407-418.

In introduction: DP-1 and DP-2 are introduced in the introduction, but these terms are not used for the explanation. To improve consistency and clarity, I recommend revising the introduction so that these terms are meaningfully connected to the analysis and discussion presented later in the paper.

We appreciate this helpful comment. In response, we expanded our data analysis to explicitly identify the DP-1 and DP-2 components following the framework of Kamide and Kokubun (1996). The data interpretation has been substantially revised to link the observed cyclic evolution of auroral currents to the DP-1 and DP-2. Correspondingly, the Introduction has been rewritten to connect these terms to the analysis and discussion. Modifications in Line 99-108,111-118.

Page 3, The term "conjugate" is used, but its meaning is unclear in this context. Do the authors mean simultaneous observations in the same region, or something else? This should be clarified to avoid ambiguity.

In our study, the term “conjugate” refers to simultaneous observations of FAC, plasma convection, and auroral electrojets within the same ionospheric auroral region. We have added its clarification in the revised manuscript. Modifications in Line 343-344.

Page 3, "Global MHD simulations provide time-resolved predictions [42, 43], but need observational validation.":

The statement is too general. I recommend specifying which aspects of the simulations require validation, for example, the location of FACs, the dynamics of the ionospheric convection, or the temporal evolution of the substorm phases. Providing concrete points will make the motivation clearer and strengthen the connection between the simulations and the observations presented in this study.

These MHD simulations shows that regions of enhanced magnetosphere convection and Region 1 FAC progress toward the nightside shortly after the southward IMF turning. These features are consistent with the longitudinal evolution of DP-2–related convection. We have incorporated these points into the revised Introduction. Modifications in Line 116-118.

Page 3, "These cycles involve coherent shifts of current peaks across magnetic local time and latitude—initially toward the nightside and equator, then reversing toward the dayside and pole.":

This statement appears inconsistent with the data presented. For the 17 March 2015 event, the current peaks are not shown.

In the revised manuscript, we have added a two-dimensional map of the SML index, using the B_n perturbations rotated by 90° to represent the westward auroral electrojet. The peak electrojet location is now explicitly marked to illustrate its temporal evolution. The corresponding descriptions have been incorporated into Figure 1. Figure 2 further presents the 12-hour evolution of the auroral-current peaks across five substorms on 17 March 2015.

Page 4, "solar wind IMF" reads as "IMF".

We correct it accordingly.

Page 4, I recommend rephrasing "nightsideward".

We have rephrased it to "antisunward"

Page 5, " Once nightside reconnection dominates, the OCB—traced by the poleward edges of Region 1 FACs and convection—retreats poleward. The peak of convection and its associated FACs subsequently propagate toward the dayside, reflecting a transition to tail reconnection–dominated convection.":

I recommend revising this sentence using more physically sound terminology rather than imaginative wording. As written, the meaning is unclear, and I am not able to fully understand the intended physical interpretation. A clearer, physics-based explanation is necessary.

We have substantially revised the relevant interpretation and relocated it to the Discussion section. In the revised analysis, we clarify that the sunward–poleward movement of the current peaks signifies an interval dominated by nightside reconnection within the OCB framework. Modifications in Lines 359-369.

Page 5, " Five distinct cycles are highlighted by blue boxes.":

What specific criterion was used to define and select these cycles? The authors should clarify the methodology or threshold employed

These cycles are identified by examining the MLT peak of SML near the expansion phase. We first search for the antisunward phase of a cycle by detecting a rapid MLT decrease exceeding $\Delta\text{MLT} > 2$ hours within 20 minutes. A subsequent increase in the MLT defines the return branch of the cycle. The resulting time windows derived from the MLT evolution are then applied consistently to the MLat panels. The cycles in MLT and MLat are not perfectly identical but are broadly coincident. Modifications are in Line 270-279.

Page 6, "all five substorm expansions occur within these cycles.":

For the second one the tendency is opposite. During the expansion phase, the peak moves to later MLTs, and during the recovery phase, it moves to earlier MLTs.

We agree with the reviewer's observation. For the second through fifth substorms, the expansion phase is mostly during the sunward-poleward phase of the cycle, which differs from the first substorm. Our interpretation is that expansion onset can occur during intervals dominated by either dayside reconnection (as in the first substorm) or nightside reconnection (as in the last four substorms). These discussions have been incorporated in Line 298-304.

Page 6, "This consistent relationship suggests that substorm expansions are modulated by the broader cyclic reconfiguration of auroral current systems."

The statement seems to lack sufficient basis. There is a significant leap from the results presented to this conclusion. The authors should provide a clearer justification or additional evidence to support this interpretation, or else rephrase the statement in a more cautious manner. The authors should also consider the possibility that the substorm activities results in the cycle of the peak locations, as Gjerloev et al. (2004) show.

In the revision, we have removed the statement that substorm expansions are modulated by the current-cycle evolution. Instead, we now discuss that the expansion phase (DP-1) itself can lead to the peak-location changes within the cycle, consistent with the findings of Gjerloev et al. (2004). This interpretation has been incorporated into the revised text.

Page 6, Figure S2, S3 and S4:

The auroral electrojets dominant, and Figure S2, S3 and S4 seem not to show such cycle clearly.

As noted in major comment #1, the challenge with previous Figures S2-S4 arose from the weak pre-expansion SML, which mostly appears dark. This made the cyclic motion difficult to see. To resolve this, we added a panel that shows the instantaneous peak SML location over time, with other MLT sectors left blank. This effectively highlights

the cyclic motion of the SML peak in MLT. This panel also reveals that the antisunward motion of the SML peak sometimes appears as discrete jumps, likely associated with the addition of nightside DP-1 currents, consistent with Gjerloev et al. (2004). We have included these clarifications and discussions in the revised manuscript.

Page 6, " Our central result is that substorm expansion is embedded within a planetary-scale cycle of auroral current systems that are tightly coupled to global plasma convection."

The statement is problematic. Numerous previous studies have already shown that substorm expansion occurs in the region where convection is present. Therefore, the novelty and physical meaning of the claim are unclear. The authors should explicitly define what is meant by "planetary-scale cycle." How does the cycle described here differ from earlier concepts, particularly the framework proposed by Kamide and Kokubun (1970, <https://doi.org/10.1029/96JA00142>)? Observations also show that some expansion onset takes place near the Harang discontinuity, which is located in the ionospheric convection (Weygard et al., 2008, <https://doi.org/10.1029/2007JA012537>). In addition, global MHD simulation results suggest that the substorm expansion is initiated and sustained in the convection system (Tanaka et al., 2010, <https://doi.org/10.1029/2009JA014676>; Ebihara and Tanaka, 2017, <https://doi.org/10.1002/2017JA024294>; Tanaka et al. (2021, <https://doi.org/10.1029/2020JA028942>). I recommend discussing these previous perspectives.

We have added substantial clarification on this point and incorporated discussions of previous perspectives.

Our focus is not on the spatial embedding of substorm expansion in a particular region. Instead, we examine the time domain, showing that the substorm expansion phase is temporally embedded within a recurrent, cyclic evolution of auroral currents. By "planetary-scale," we specifically mean the global longitudinal progression of current peaks across MLT. In the revised manuscript, we explicitly link this cyclic evolution to the DP-1 and DP-2 current systems in the Kamide and Kokubun (1996) framework. These cycles provide new constraints on the relative timing of DP-1 and DP-2 and their connection to intervals dominated by either dayside or nightside reconnection.

We have also expanded the discussion of earlier studies on convection. Weygard et al. (2008), Ebihara and Tanaka (2017), and Tanaka et al. (2021) describe localized convection flows directly associated with DP-1 onset, whereas Tanaka et al. (2010) emphasize the global DP-2–driven convection,. Our analysis focuses on the temporal relationship between this global DP-2 convection and the DP-1 current development, as well as their coordinated evolution. These points have now been incorporated into Lines 379-400, 418-422.

Page 7, " dominant dayside reconnection opens magnetic flux, expanding the polar cap and pushing the OCB equatorward."

What do you mean by "dominant"? Do you mean that if the dayside reconnection is minor, the polar cap location is unchanged?

By "dominant dayside reconnection," we mean that the dayside reconnection rate significantly exceeds the nightside reconnection rate. In this case, closed magnetic flux is converted into open flux faster than it is closed on the nightside, leading to an expansion of the polar cap and an equatorward shift of the OCB. If the dayside reconnection rate is small, or comparable to the nightside reconnection rate, the total open-flux content changes only slightly and the polar cap boundary remains nearly stationary. These clarifications are consistent with the ECPC framework in lines 359-360.

Page 7, "closing magnetic flux and contracting the polar cap, leading to a poleward retreat of OCB, auroral currents and convection. Correspondingly, nightside-driven convection and Region 1 FAC progress toward the dayside."

Please explain the reason why contracting polar cap results in poleward retreat of auroral currents and convection? The convection flows entirely at all magnetic latitudes from the magnetic pole to the magnetic equator.

In this study, we examine sunward convection within the closed-field auroral region, distinct from the low-latitude convection or high-latitude convection in the polar cap. Region 1 FACs and the sunward return flow of the two-cell convection are associated with the low-latitude boundary layer (LLBL; e.g., Sonnerup, 1980). The strongest Region 1 FACs map near the LLBL, while the sunward return flow is immediately equatorward of LLBL. The DP-2 auroral electrojet is also directly coupled to this sunward convection. As the polar cap contracts, the open-closed boundary (OCB) shifts poleward, and the LLBL follows. So, the associated auroral currents and the sunward convection channel are expected to move poleward together. The co-moving of Region 1 FAC and the convection are consistent with observations (Fogg et al., 2020JGR, Walach et al., 2025JGR).

Page 7, "the first substorm expansion occurs within convection driven by dominant dayside reconnection.":

Is this statement based on evidence, or speculation?

The statement is based on observational evidence, supported by previously modeling.

For the first substorm, the expansion phase coincides with (i) an antisunward and equatorward movement of the peak Region 1/DP-2 auroral electrojet, and (ii) an enhancement of global convection in the 03–09 MLT sector observed by SuperDARN. The antisunward progression of Region 1 FACs has been identified as a signal of convection driven by dominant dayside reconnection (Dai et al., 2024; Zhu et al., 2024, 2025). Likewise, the equatorward shift of FACs and the auroral electrojet is consistent

with polar cap expansion during a dominant dayside reconnection. Taken together, these observational signatures support the proposed physical interpretations.

Page 7, "Such substorm is likely directly coupled to real-time solar wind drivers":

I recommend revising this statement. What do you mean by real-time?

We have revised the statement to avoid the ambiguous term "real-time". Our intended point is that, the growth and decay of the first substorm are closely tied to temporal variations in solar wind driving. Specifically, enhanced southward IMF Bz and Ey correspond to the substorm intensification, while a northward turning of IMF Bz coincides with the recovery phase for this substorm. We have clarified this interpretation in the revised manuscript. Modifications in Line 307-310.

Page 7, "the four subsequent substorm expansions take place during convection when nightside reconnection dominates.":

Is this statement based on evidence, or speculation?

The statement is based on observational evidence, supported by previously established modeling and theoretical frameworks.

Observationally, these substorm expansions take place primarily during the sunward-poleward movement of the FAC and auroral electrojet peaks. This interval is accompanied by enhanced global convection in MLT = 3–9, as documented by SuperDARN. Conceptually, the poleward motion of the FAC and auroral electrojet is associated with contraction of the polar cap, which corresponds to dominant nightside reconnection (Cowley and Lockwood, 1992; Milan and Grocott, 2021). We have revised the statement to clarify this point. Modifications in Line 359-364.

Page 8, "unified framework in which reconnection-driven global convection modulates substorm expansion, while mesoscale processes shape its onset timing and substructure.":

The statement lacks a clear basis. It is already well established that substorm expansion takes place within the large-scale convection system. The authors should clarify what new evidence supports this claim, and specify the major difference between their proposed framework and existing knowledge.

We have revised the discussion accordingly. The original statement on "modulation" has been replaced with an interpretation directly grounded in our observations. In the revision, we incorporate an analysis of the DP-1 and DP-2 current systems, which provides new constraints on the timing of DP-1 and DP-2 relative to the phases of the cyclic evolution of auroral currents. This represents an advance in addition to the identification of the cyclic evolution. Modifications in Line 379-400.

Page 8, "but part of a global reconnection-modulated convection cycle that reconfigures auroral current systems":

Same as above.

Thank you for this comment. We have modified the interpretation and discussion

session. This statement has been replaced with interpretation that is more directly based on observations in this study.

Page 9, "Region 1 FACs originate from the low-latitude magnetospheric boundary layer [54], and typically map to sunward convection flows on closed field lines [55, 56]."

Do [55, 56] actually explain why FACs originating from the boundary are mapped into the closed region? In those studies, the assumption is that FACs propagate strictly along magnetic field lines, implying that boundary-originating FACs should map to the OCB in the ionosphere, rather than closed regions.

In the revision, we clarify the mapping of region 1 FACs to closed field lines and provide more accurate references.

Reference [56] (Sonnerup, 1980) presents a low-latitude boundary layer (LLBL) model in which Region 1 FACs can extend into the closed field line region. In this framework, the FAC magnitude is proportional to the flow shear in the magnetospheric convection. This conceptual model is illustrated in the enclosed Figure R2. In addition, our global magnetospheric simulations (Zhu et al., 2024, 2025; Dai et al., 2024) are consistent with this picture: we observe sunward convection and Region 1 FACs extending into the closed magnetosphere, rather than being confined strictly to the open-closed boundary. These results support the idea that Region 1 FACs can map into the closed region under realistic convection patterns. Modifications in line 364-369.

Fig. 1. View from the sun of the morningside boundary layer and its coupling to the ionosphere. Regions 1 and 2 correspond to the field-aligned current regions of Iijima and Potemra [1976a]. The constant equatorial current j_{00} is deflected either eastward or westward into the ring current at the inner edge of region 2.

Figure R2. Reproduction of Figure 1 from Sonnerup (1980). This schematic illustrates that the Region 1 FAC can map into both the low-latitude boundary layer (LLBL) and the sunward convection region. The FAC magnitude is proportional to the flow shear within the convection system.

Page 10, Equation (1):

This equation includes time derivative. Is it possible to explain the Region 1 FACs in which the time variation is not so high.

Equation (1) is derived from first principles and describes the fundamental relationship between flow shear and FAC generation. Even for Region 1 FACs that vary slowly in time, this formulation remains valid. Alternative formulations, such as those in Ebihara and Tanaka (2022, JGR), also relate FAC generation to the electric field and retain a time-derivative term, consistent with the first-principles approach.

Page 10, "Conceptually, this expression shows that Region 1 FAC enhancements tend to coincide with regions of intensified convection.":

The convection is the circular motion of plasma in the magnetosphere. Do you intend to mean that the Region 1 FACs are generated in the magnetosphere, rather than the boundary layer? [56] considers flow shear around the boundary to cause the Region 1 FACs.

According to Sonnerup (1980) [56], Region 1 FACs can be generated both in the low-latitude magnetospheric boundary layer and within the closed magnetosphere closer to Earth, as illustrated in enclosed Figure R1. Enhanced convection within the closed magnetosphere can contribute to FAC generation through flow shear. This scenario is also supported by global simulations (Zhu et al., 2024, 2025; Dai et al., 2024). The mathematical relationships linking Region 1 FACs to convection are detailed in references [35, 54, 55]. In the revised manuscript, we have clarified these points. Modifications in line 364-369.

Page 10, "Strengthening magnetospheric convection enhances the ionospheric convection electric field and increases large-scale FACs, which serve as a proxy for auroral precipitation.":

Do you mean that magnetospheric convection, not plasma flows at boundary, enhances Region 1 field-aligned currents (FACs), or the ionospheric convection enhances Region 1 FACs? What is the relation to the southward component of IMF? How is this related to auroral precipitation? As I understand it, the authors are focusing on the dawnside Region 1 downward FACs. In the downward FAC region, significant auroral precipitation is generally not expected.

Consistent with previous studies [35, 55, 56] and global MHD simulations (Dai et al., 2024; Zhu et al., 2024, 2025), large-scale Region 1 FACs can be driven by magnetospheric sunward convection. The convection and associated FACs are directly enhanced by increases in the southward IMF Bz (Dai et al., 2024). We agree with the

comment on the auroral precipitation. Accordingly, we have removed the statement linking FACs directly to auroral precipitation in the revised manuscript.

Page 10, "The AMPERE field-aligned current is available on <http://ampere.jhuapl.edu/rBrowse/index.html>":

This site is inaccessible as of September 1.

The modified website is <https://ampere.jhuapl.edu/browse/>

Page 10, " SPEDAS codes used for analyzing THEMIS data are freely available at <http://spedas.org/blog/>":

The site is also inaccessible as of September 1.

The updated site for SPEDAS codes is <https://themis.igpp.ucla.edu/software.shtml>

We sincerely appreciate your feedback, which has been invaluable in guiding our revision. In response, we have expanded data analysis, clarified key points, and refined our interpretations. Detailed, point-by-point responses are provided below.

REVIEWER COMMENTS

Reviewer #2 (Remarks to the Author):

Review of “Substorm Expansion Embedded in a Planetary-Scale Auroral Current Cycle”

This paper presents observations of auroral current systems and ionospheric convection, alongside indicators of magnetospheric substorm activity. The auroral current systems and convection data provide supporting evidence for the expanding contracting polar cap (ECPC) paradigm. In brief, the ECPC states that intervals of dominant low-latitude dayside reconnection enhance currents and flows on the dayside, leading to polar cap expansion, and intervals of dominant nightside reconnection enhance currents and flows on the nightside and lead to polar cap contraction. Low-latitude dayside reconnection is driven by coupling with a southward IMF, and nightside reconnection is commonly expected during substorms. Which of these is dominant is typically dependent on the magnitude of negative IMF B_z and the magnitude of any concurrent substorm activity.

The first event identified in this study has a substorm occurring during strongly southward IMF. As such, dayside reconnection is observed to be dominant despite the occurrence of the substorm. In subsequent substorms, nightside reconnection is observed to be dominant. It is claimed that these results “uncover a reconnection-driven convection process that modulates substorm expansion on planetary scales”. This seems to be a rather inflated claim. The ECPC paradigm fully predicts the reported behaviour. Whilst the observations presented here are consistent with expectations, they do not uncover anything unexpected.

We agree that the ECPC paradigm captures the latitudinal expansion–contraction evolution observed in these cycles. However, our observations reveal features not previously documented in the ECPC framework.

First, we identify a coherent longitudinal (MLT) cyclic evolution of auroral currents. To our knowledge, such MLT evolution has not been described or discussed in the ECPC literature (Cowley and Lockwood 1992; Milan and Grocott 2021). For instance, while ECPC predicts enhanced dayside currents during intervals of dominant dayside reconnection, it does not specify motion of the current peak across local-time sectors; in principle, these enhanced currents could remain fixed on the dayside.

Second, the revised manuscript incorporates timing analyses of DP-1-like and DP-2-like substorm currents through the cycles. These observations provide a connection between the ECPC framework and DP-1-DP-2 framework (e.g., Kamide and Kokubun

1996; Akasofu 2017).

These features are now highlighted in our revised discussion to clarify the observational advances with respect to the established ECPC context. Modifications in Line 369-377, 379-400. We have also removed language (e.g., “uncover a reconnection-driven convection process”) that appears as overstating the scope of our inference.

It is not clear what the “planetary” scale modulation of substorms is. There is not really any evidence provided that the reconnection cycle is “modulating” substorm expansion. It is expected by the ECPC paradigm that substorms will modulate the currents and convection, as shown here (this has of course been well-reported)..

We agree that the previous wording was ambiguous. In the revision, we clarify that “*planetary scale*” refers specifically to the global MLT extent of the auroral current cycle. Our interpretation now focuses on the longitudinal (MLT) cycle and the timing of substorm expansion within the MLT–Mlat cycle. We have removed the previous use of “*modulation*”, which implied causality.

Perhaps the substorm expansion itself could be modulated by the concurrent upstream conditions and resulting magnetospheric dynamics. The MLT and MLAT of each substorm is different, for instance, and further analysis could determine whether those differences had any relation to the ongoing large-scale dynamics. But this would not necessarily be a new result – previous work (e.g. Milan et al., 2009, doi:10.5194/angeo-27-659-2009; Milan et al., 2010, doi:10.1029/2010JA015663) has looked at how the substorm expansion phase evolves under different solar wind driving conditions. That later substorms occur during the ensuing main phase of the geomagnetic storm, with its associated ring current enhancement, is also not considered, but has also been shown previously to influence the evolution of the substorm (Milan et al., 2008, doi:10.1029/2008JA013340).

We appreciate the perspectives outlined in this comment. The upstream solar wind conditions and ring current dynamics may provide insights into the modulation of substorms. Consistent with our response to the previous comment, we have removed wording that implied causal ‘modulation’ of substorms.

Milan et al.,2009 shows that the substorm intensity is governed by the open flux content, which could relate to enhanced dayside reconnection and enhanced ring current. Milan et al.,2010 shows that auroral bulge expands eastward and westward with respect to the substorm onset MLT during the expansion phase, consistent with the DP-1 electrojet evolution [Kamide and Kokubun 1996]. Milan et al. (2008) shows that an intensified ring current can make the magnetotail more resistant to reconnection. This provides a natural explanation for the delay between the interval of strong solar-wind driving (i.e., dominant dayside reconnection) and the onset of the expansion phase in the later substorms when SYM-H is depressed to -100nT ~-

200 nT. These references have been incorporated into the discussion of revised manuscript. Line 315-325,394.

Overall, although the observations provide a nice example of a well-established phenomenon, there is insufficient detail in the analysis or discussion to establish what genuine advancement to the field these observations might offer.

We appreciate your recognition of the technical soundness of our observations. Following your suggestion, we have expanded the analysis and discussion to clarify the study's contributions relative to the established ECPC framework in the revised manuscript.

First, we show that the auroral current system exhibits a longitudinal (MLT) cyclic evolution, adding spatio-temporal dynamics that ECPC does not address. Second, we identify when DP-1 like substorm currents occur within these cycles, providing information that ECPC does not specify. These points provide substantive advances beyond existing ECPC-based descriptions and are now explicitly articulated in the revised discussions.

Minor comments:

On page 4 (section 2.1) it is stated that "Figure 1 presents the spatial and temporal development of Region 1 FACs (blue) and sunward ionospheric flows in the dawn sector." Figure 1 presents a global view of the FACs and convection. The FACs are not only shown in blue but also red. The full figure should be introduced before then focussing on a particular aspect.

We have revised the description of Figure 1 to introduce the complete figure before discussing specific aspects

In reference to Figure 2: It would be useful to see the IMF Bz component throughout, to assess whether the difference between the "descending" nature of the first substorm and the "ascending" nature of the subsequent ones was related to the concurrent solar wind conditions. Although Bz is shown for a wider interval in the SI, it is not clear. In fact, it might be more useful to present in Fig. 2 the solar wind electric field, which is expected to more directly correlate with the strength of dayside reconnection.

We have added IMF Bz and the solar wind electric field E_y to Figure 2. These additional observations are consistent with previous discussions. The first substorm appears to be directly driven by the solar wind: its growth and expansion phases coincide with enhanced E_y and southward Bz, while its recovery phase is con-incident with northward turning of IMF Bz and negative E_y . For the last four substorms, the expansion phases occur during either constant or decreasing E_y , consistent with that their expansion is primarily related to dominant nightside reconnection as regulated by internal magnetospheric processes. In the revised manuscript, the discussion has been updated to incorporate these solar wind electric field observations.

Modifications in Line 307-315.

Section 4.3: What statistical fitting technique was used to assimilate the SuperDARN data? A commonly used fitting technique is that described by Ruohoniemi and Baker (1998, doi:10.1029/98JA01288) but this is not referenced. If this is the method used, then further details of the fitting process (radar data gridding options, IMF input, background model, HMB identification etc.), need to be provided for reproducibility.

We add references and more details in processing the SuperDARN data. The processing of SuperDARN data follows the standard procedures implemented in the Radar Software Toolkit (RST v5.0; <https://radar-software-toolkit-rst.readthedocs.io/en/latest/>), utilizing the Map-Potential fitting technique as described by Ruohoniemi and Baker (1998). The detailed procedures are added in the method section 4.3. Modifications in Line 473-490.

We sincerely appreciate your feedback, which has been invaluable in guiding our revision. In response, we have expanded data analysis, clarified key points, and refined our interpretations. Detailed, point-by-point responses are provided below.

REVIEWER COMMENTS

Reviewer #3 (Remarks to the Author):

Key Results: This manuscript explores the relationship between substorm expansion phases and large-scale plasma convection cycles. Sharp decreases in the AL index identify the substorm expansion phases, while convection cycles are determined using AMPERE-derived pictures of the field-aligned currents (FACs), as well as Super-DARN-derived convection maps. The authors conduct an event study using five substorms that occurred during the 17 March geomagnetic storm. Initially, they investigate the first of the five substorms. They demonstrate that during the expansion phase of this substorm, the region-1 (R1) FACs and convection signatures migrate equatorward (Fig. 1). At the same time, their peak intensities shift nightward from the pre-noon sector to the post-midnight sector. This is interpreted, following the expanding/contracting polar cap (ECPC) model, as an equatorward expansion of the open-closed boundary (OCB), implying that dayside/magnetopause reconnection is the dominant driver of magnetospheric convection during this interval. Thus, the resulting convection proceeds from the dayside towards the nightside. About ten minutes later, during the late expansion and recovery phases of the substorm, the inverse behavior is observed, with the FACs and convective flows moving poleward and shifting back toward the dayside, implying a contracting OCB, with nightside/tail reconnection driving the convection patterns.

The next section of the manuscript builds upon this analysis and extends it to all five substorms. To further illustrate the pattern of convection cycles, the authors plot the time series of the location of the peak intensities of the R1 FACs and the SML index (a metric for the strength of the westward auroral electrojet commonly attributed as the ionospheric portion of the substorm current wedge) along with the convection strength (Fig. 2). They demonstrate that each substorm expansion is embedded within a longer (~1–2 hours) convection cycle. The peaks in the FACs and westward auroral electrojets consistently shift nightward/sunward and equatorward/poleward during the “descending”/“ascending” phase of these convection cycles. The “descending” phase of the convection cycle equates to the dayside-dominated convection and the “ascending” to nightside-dominated, as is summarized in Fig. 3. Interestingly, the first substorm expansion phase occurs during the “descending” phase of the convection cycle. In contrast, the following four all occur during the “ascending” phase. The authors contend that the first substorm is likely directly driven, as described by Dai et al. (2023), while the remaining substorms result from nightside/tail reconnection.

Validity: My critique of this study hinges on the fact that the five substorms considered occur during a strong geomagnetic storm, indeed, the strongest storm of Solar Cycle 24. As strong storms are generated by abnormally strong geomagnetic driving, the results of this study may not be generalizable to all traditional and non-storm-time substorms.

Akasofu's (1964) original description of substorm onset included a sudden brightening of the pre-onset auroral arc, followed by a localized auroral bulge forming near midnight local time that expands poleward and azimuthally as the expansion phase progresses. Although it is commonplace in the community to now identify substorm onsets using sudden decreases in the AL or SML indices, as done in this study, this approach may identify substorms that differ in character from the traditional auroral picture, especially during storms. The strong and often long-duration solar wind driving muddies the situation, making it difficult to differentiate between the various substorm phases and other substorm-related phenomena such as pseudobreakups and steady magnetospheric convection (SMC) intervals. For example, the second shaded segment in Fig. 1, described as the recovery phase of the first substorm, could be interpreted as an SMC interval. Walach & Milan (2015) argue that most SMC intervals are simply substorms that continue to be driven throughout what would otherwise be their expansion and recovery phases. This has led some researchers to introduce a new classification termed "multiple intensifications" (Milan et al., 2021, 2024). Using this new classification, Bower et al. (2025) expanded upon the Substorm Onsets and Phases from Indices of the Electrojet (SOPHIE) algorithm (Forsyth et al., 2015) to include multiple intensifications. Termed SOPHIE-M, this algorithm labeled all five substorms identified in this study as multiple intensifications. Further complicating our understanding of storm-time substorms, Zou et al. (2025) recently demonstrated that very intense storm-time substorms may not be traditional substorms at all, but rather global-scale convection events distinct from traditional substorms.

These critiques do not negate the primary results and interpretation of the study. Substorms, SMCs, and multiple intensifications are not inherently distinct modes. Instead, in my opinion, they represent a continuum of substorm-like processes in the magnetosphere. The distinction between when a recovery phase is called an SMC and when numerous expansion phases are labeled as multiple intensifications is arbitrary. However, more care should be taken to avoid generalizing the results to all traditional and non-storm-time substorms. For example, in the abstract, the sentence "Based on coordinate observations of a series of intense substorms..." should be "Based on coordinate observations of a series of intense storm-time substorms...". Furthermore, the Discussion section should outline some of the study's limitations, for example, in relying on auroral electrojet indices to determine substorm expansion rather than using auroral imaging. The authors could also expand upon the description of the three other intervals included in the Supplementary Information. Currently, they are briefly mentioned in a single sentence. Fig. S2 and S3 are both during non-storm intervals, and many of the highlighted substorms appear to be simpler and more akin to the

traditional substorm description based on the AL profiles. Simply stating that the Supplementary Information contains two intervals of non-storm-time substorms would justify the generalizability of the main text's findings.

Significance: Substorm expansion phases are among the most significant magnetospheric space weather events, resulting in an explosive global-scale reconfiguration of the magnetosphere on timescales of tens of minutes. Their linkage to the ionosphere produces intense aurorae, ionospheric currents, and perturbations in the ground magnetic field. As such, understanding the physical mechanisms driving substorms is critical to eventual space weather modeling and prediction. As the authors state in the introduction, much of substorm research has focused on exploring substorm onset mechanisms and the micro-scale details of reconnection. Meanwhile, relatively little attention has been devoted to understanding the global-scale mechanisms and effects of substorms, particularly during the expansion phase. This is mainly because in-situ spacecraft observations, such as those from MMS and THEMIS, only provide a micro-scale view of the global magnetosphere-ionosphere (MI) system. However, global-scale datasets do exist but have been underutilized by the scientific community; for example, the AMPERE-derived pictures of the FACs, collections of ground-based magnetometers capable of resolving the impact of ionospheric currents, which are distilled into geomagnetic indices, and SuperDARN-derived ionospheric convection maps. Therefore, studies like this address a gap in our scientific understanding of substorms and help contextualize micro-scale investigations.

Data and Methodology: The data and methodology are sound. All the datasets used in this study are widely employed by the scientific community and readily accessible. The methodology used is straightforward, easy to understand, and is described in sufficient detail to support replication of the results. The proper links to the data and source code are also provided.

Analytical Approach: The study relies on coordinated data observations and does not employ statistical or other analytical approaches. As such, this section of the review does not apply to this manuscript.

We appreciate your careful assessment and agree with the points raised in the validity section. We agree that substorms, SMCs, and multiple intensifications likely form a continuum of substorm-type activity. The substorms in our events share characteristics with the multiple intensifications (Milan et al., 2021) identified by the SOPHIE-M algorithm (Forsyth et al., 2015, Bower et al., 2025), and with global-convection events (Walach & Milan, 2015; Zou et al., 2025). The key distinction, however, is that multiple intensifications are typically marked by a lack of coherent expanding–contracting motion (Milan et al., 2021), whereas substorm in our study exhibits clear latitude evolution of aurora currents. Regarding the distinction with purely-convection, SMU amplitudes only reach up to roughly half of SML in several

events (Fig.S1)--suggesting that convection is significant, though SML is not purely a measure of convection. Modifications in Line 326-338.

In the revision, we replaced AL with SML for identifying substorm phases. This change increases the total number of substorms in the interval to six. For the second substorm, the AL index is small because the perturbations associated with SML is at lower latitude than AL station (Figure 2d). We also added a discussion of the limitations associated with identifying expansion phases using SML rather than auroral observations. Modifications in Line 400-404.

In the revision, we have explicitly integrated these discussions. We also specify in the abstract that the events analyzed are intense storm-time substorms, and we clarify that the Supplementary Information contains two intervals of non-storm-time substorms. Modifications in Line 40, 354-357.

Suggested Improvements:

My major suggestions have been discussed in the Validity section above. Some minor suggestions are:

Fig. 1: It would be helpful to add a horizontal line in panel (a) at $B_z = 0$ to distinguish when the IMF is southward vs. northward.

We have added the horizontal line at $B_z=0$ in Figure 1 and 2.

Fig. 2: To understand the context of convection cycles and substorm expansions in relation to solar wind driving, it would be helpful to add a panel showing the IMF B_z , similar to Fig. 1a. If this makes the figure too large, panels (b) and (c), as well as panels (d) and (e), could be combined by overplotting the yellow lines overtop of the bars. This may also make the correlations between these sets of panels more evident.

We have followed the suggestion and add a panel of IMF B_z and E_y in Figure 2. In addition, we combine the panels by overplotting the lines overtop of the bars.

Fig. S2–S4: To understand the storm context of these substorm expansions and convection cycles, it would be helpful to add additional panels showing the Sym-H index similar to Fig. S1c.

We have added the SYM-H index in the Fig.S2-S4.

Clarity and Context: The manuscript is well-written, clear, and develops in a logical and readily understandable fashion. The introduction is concise yet nicely frames the study within its historical and scientific context.

References: The manuscript provides ample references, several of which I consulted during the evaluation of this manuscript.

Your Expertise: The reviewer is an expert in empirical magnetic field modeling of planetary magnetospheres. In particular, the reviewer has used flexible empirical magnetic field models in conjunction with data mining approaches to resolve global-scale magnetospheric dynamics during storms and substorms.

We thank all the reviewers for the careful and constructive comments. These feedbacks have been very helpful in improving the clarity and interpretation of the results. We have revised the manuscript accordingly and address each comment in detail below.

REVIEWER COMMENTS

Reviewer #1 (Remarks to the Author):

[Major comments]

My first major concern in the previous round was the supporting evidence for the proposed cycle. I acknowledge that Figure 1 has been substantially improved, and it is now possible to identify the evolution of the FACs more clearly. The inclusion of equivalent current maps is also helpful. I agree that the Region 1 FACs and the westward auroral electrojet exhibit a coherent antisunward and equatorward progression during the expansion phase. However, an alternative interpretation remains possible: The Region 1 FACs and the auroral electrojet may develop discretely rather than continuously. In Figures 1c and 1d, the dayside FACs remain strong (as highlighted by the green circle), while enhanced nightside FACs appear separately (as indicated by the grey arrow). This behavior suggests the formation of distinct current systems, rather than a smooth, continuous equatorward and antisunward migration. Therefore, I am not fully convinced that the FAC and auroral electrojet peaks move equatorward and antisunward in a smooth manner, as schematically depicted in Figure 3. Additional clarification or quantitative evidence would be necessary to distinguish between continuous motion and the discrete development of multiple current systems. Figure 2b, which shows the MLT variation of the auroral electrojet, also exhibits discrete changes, rather than a smooth evolution in MLT. If the evolution of the FACs and the auroral electrojets is indeed discrete, this interpretation would be consistent with the authors' revised description (Lines 169–175). In that case, however, the novelty of the proposed "cycle" needs to be clarified more explicitly, particularly in relation to previously reported discrete developments of substorm-related current systems. If a smooth migration cannot be clearly demonstrated, the applicability of the ECPC framework would become unclear, and a more careful and internally consistent physical explanation would be required.

We appreciate this careful and constructive comment.

First, we agree that the peaks of the currents do not necessarily evolve continuously and can develop in a discrete, stepwise manner. This behavior is particularly evident during the first two substorms. In the revision, we explicitly clarify and emphasize this point. Figure 3 has been modified to

incorporate stepwise motion in MLT, and the text has been revised accordingly. In the Discussion we now state that “the longitudinal motion of the current peaks.....appears more stepwise in certain intervals (e.g., Fig.~1), consistent with discrete enhancement and decay of nightside DP-1 currents (Gjerloev et al. 2004).” (line 352–356)

At the same time, a careful re-examination of the observations shows that the longitudinal motion of the current peaks can also appear predominantly smooth. This behavior is clearly illustrated during the sixth substorm (fifth cycle), in added Fig.S3. In the revised Results section, we therefore add: “For the fifth cycle as in supplementary Fig.S3, the motion of the current peaks appears mainly smooth, in contrast to the more stepwise progression observed during the first substorm (Fig.1)”. (line 231-238). Quantitative evidence supporting this interpretation is now included in Supplementary Fig. S3, and the pattern of this interval is consistent with Fig. 2d.

Based on these clarifications, we refine the interpretation to emphasize that both stepwise and smooth progressions can occur. This reflects the fact that DP-1 and DP-2 currents coexist during substorms, with their relative contributions varying in time. Accordingly, we revise the summary to state that “The longitudinal motion of the current peaks exhibits a mixed nature, combining stepwise and smooth progression: it appears more stepwise in certain intervals (e.g., Fig.~1), consistent with discrete enhancement and decay of nightside DP-1 currents (Gjerloev et al., 2004), and smoother during others (e.g., Fig.~S3), likely reflecting the continuous evolution of convection-driven DP-2 (Milan,2013, Dai et al.,2024).” (Line 350-357).

This refined interpretation is also incorporated into the updated schematic and caption of Fig. 3, which now explicitly indicates that the progression of current peaks can involve stepwise or smooth motion.

Fig.S3, Region~1 FAC and local SML () during fifth cycle (6th substorm), showing that the longitudinal evolution of current peaks is more continuous in this cycle.

[Minor comments]

I raised a concern about the use of the term "planetary-scale auroral current." In their response, the authors state: "In this study, the phrase 'planetary-scale' denotes the global extent of the observed current cycle across a wide range of magnetic local times. The phrase "auroral currents" denotes the combined system of auroral electrojets within the ionosphere and the auroral field-aligned currents (FACs) that connect the ionosphere to the magnetosphere. Auroral electrojets are widely considered a component

of the auroral current system, and the term 'auroral FACs' has also been used in previous studies (e.g., Lysak, 1985; Elphic et al., 1998; Juusola et al., 2016; Xiong et al., 2021). To avoid ambiguity, we have clarified these definitions in the Introduction." the term is not well clarified in the Introduction section. Aurora refers to an optical phenomenon, whereas current is a physical quantity describing the flow of electric charge per unit time. The physical connection between aurora and current is therefore not self-evident in the present paper and requires careful clarification. While some previous studies (e.g., Lysak and Elphic) have used the term "auroral FACs," it is clear in those works that the intention was to describe FACs associated with electron precipitation. In contrast, the present paper focuses primarily on downward FACs, in which the relation between FACs and field-aligned currents are unclear. This distinction is probably important and is not sufficiently addressed. Moreover, it is well known that both FACs and the ionospheric equivalent currents (e.g., DP2) can exhibit global-scale structures. In this context, introducing the additional qualifier "planetary-scale" does not seem necessary, nor does it add clear physical meaning. In fact, the term "planetary-scale" may be misleading, as it could be interpreted as implying processes confined near the planet itself, rather than processes involving the magnetosphere as a whole. For these reasons, I again recommend rephrasing or avoiding the term "planetary-scale auroral current", and instead using terminology that more precisely reflects the physical quantities and processes actually analyzed in this study.

We agree with this comment and have revised the manuscript accordingly. To avoid ambiguity and better reflect the physical quantities analyzed, we have removed the terms "planetary-scale" and "auroral currents" throughout the manuscript. The former has been replaced by "global-scale", and the latter by explicit references to field-aligned currents (FACs) and auroral electrojets (AEJs).

Consistent with this change, the title has been revised to "Substorm Expansion Embedded in a Global Cycle of Field-Aligned Currents and Auroral Electrojets."

In the Introduction, we now explicitly define the relevant current systems in physical terms: "large-scale ionospheric and field-aligned currents (FACs). These include the eastward and westward auroral electrojets (AEJs) in the auroral-zone ionosphere and the FACs coupling the ionosphere to the magnetosphere." (Line 93-96)

Line 116-118: "Global simulations show that convection-driven Region 1 FACs develop from the dayside toward the nightside [28, 32–34],...":

I recommend rephrasing "convection-driven" by "convection-associated" because [28, 32-34] seems not to state explicitly that the convection drives the Region 1 FACs.

We agree with this comment and have revised the phrasing to refer to "convection-associated" Region 1 FACs. Line 116

Reviewer #2 (Remarks to the Author):

The authors have thoroughly addressed most of the comments raised in the first set of reviews. It is much improved, but on one point I am still not satisfied the authors have fully considered the implications of the existing literature.

In their response to my review they state "we identify a coherent longitudinal (MLT) cyclic evolution of auroral currents. To our knowledge, such MLT evolution has not been described or discussed in the ECPC literature" and "we show that the auroral current system exhibits a longitudinal (MLT) cyclic evolution, adding spatio-temporal dynamics that ECPC does not address".

In the revised manuscript they write: "Notably, the cycles display a coherent longitudinal (MLT) movement of the current peaks—an antisunward motion during the equatorward phase and a sunward return during the poleward phase. This longitudinal evolution of currents has not been reported and discussed in previous ECPC studies".

It is implicit in the ECPC paradigm that the foci of the convection cells (and hence the peaks of the FAC) evolve in local time as the dayside (solar wind driven) and nightside (substorm driven) reconnection dominance varies. When magnetopause reconnection is dominant the convection enhancement, that begins on the dayside, then expands antisunward, whilst the polar cap also expands. When nightside reconnection is dominant the convection enhancement, that begins on the nightside, then expands sunwards, whilst the polar cap contracts. This is exactly what is described by the authors, above.

Studies of a local time evolution of the FAC (or convection cells) are not new. Lockwood et al. (1986) provided some of the earliest evidence of flows which "after a southward turning of the IMF were propagated... around the afternoon sector". More recently, Milan (2018) modelled the evolution of the FAC through cycles of the ECPC paradigm. His model "can be used gainfully to understand the factors that determine region 1 and 2 current intensities and, most importantly, the dynamics of the current systems under different

solar wind-magnetosphere coupling conditions. These include changes in the latitude of the current regions as the polar caps expand and contract and the local time of the current maxima as dayside and nightside reconnection rates vary.”

As such, I think it would be appropriate to acknowledge how the observations presented in this paper – including an MLT evolution – can in fact be interpreted in terms of the ECPC paradigm. That is not to say that the other interpretations offered by the authors, such as the varying dominance of the DP1 / DP2 systems, should not also be given due consideration. That said, it is probably also important to note that the DP1 / DP2 classification of the ionospheric current systems is not incompatible with the ECPC paradigm either (e.g. Milan et al., 2017).

References:

Milan, S. E. (2013), Modeling Birkeland currents in the expanding/contracting polar cap paradigm, *J. Geophys. Res. Space Physics*, 118, 5532–5542, doi:10.1002/jgra.50393.

Lockwood, M. et al. (1986), Eastward propagation of a plasma convection enhancement following a southward turning of the interplanetary magnetic field, *Geophys. Res. Lett.*, 13, 72-75.

Milan, S. E. et al. (2017), Overview of Solar Wind–Magnetosphere–Ionosphere–Atmosphere Coupling and the Generation of Magnetospheric Currents, *Space Sci. Rev.* 206:547–573 doi:10.1007/s11214-017-0333-0.

We appreciated this careful and constructive comment.

We fully agree that longitudinal motion of currents and convection has been implied in previous ECPC studies (Lockwood et al., 1986; Milan, 2013). In the revised manuscript, we have clarified this point in the Discussion section (Line 379-387): “Previous ECPC studies also showed longitudinal evolution of convection and currents (Lockwood et al.,1986, Milan,2013, Dai et al, 2024). In particular, antisunward propagation of enhanced ionospheric convection following southward IMF turnings has been observed (Lockwood et al.,1986, Dai et al, 2024}. ECPC-based modeling of Region~1 FACs (excluding DP-1–related currents) shows local-time evolution of current maxima as the balance between dayside and nightside reconnection varies (Milan,2013).”

We also confirm that the DP-1/DP-2 paradigm is consistent with the ECPC paradigm. The mostly smooth local-time evolution reported in previous ECPC studies likely corresponds to convection-driven DP-2 currents. We clarify this

further in the Discussion:

“To understand the longitudinal evolution, we classify the AEJ into DP-1 and DP-2 components following Kamide et al. (1996), a classification fully compatible with the ECPC paradigm (Milan et al., 2017).” (Line 391-392)

“The longitudinal motion of the current peaks exhibits a mixed nature, combining stepwise and smooth progression: it appears more stepwise in certain intervals (e.g., Fig.~1), consistent with discrete enhancement and decay of nightside DP-1 currents (Gjerloev et al., 2004), and smoother during others (e.g., Fig.~S3), likely reflecting the continuous evolution of convection-driven DP-2 (Milan,2013, Dai et al.,2024).” (line 350-357)

“Taken together, these observations establish an explicit connection between the ECPC paradigm and the DP-1/DP-2 paradigm.” (Line 415-419)

Reviewer #3 (Remarks to the Author):

The authors have made considerable efforts to adequately address all my major and minor comments from the previous review and, in doing so, have significantly improved the clarity of the manuscript. Likewise, the authors have made significant changes to address the other reviewer's comments. As such, I recommend that the manuscript be accepted for publication in its current form.

Thank you for your positive assessment. We greatly appreciate your evaluation and are glad that the revisions have improved the clarity and quality of the manuscript.

Reviewer #3 (Remarks on code availability):

The source code is provided in a Zenodo archive, along with a brief README file that describes the source code and datasets. Overall, the analysis in this study is relatively simple, relying on coordinated data observations, as reflected in the code's simplicity. The majority of the code is dedicated to generating figures from standard, widely used datasets. Although I did not attempt to run any of the code myself, replicating the figures from the code seems straightforward.